# Federated Learning with Convex Global and Local Constraints

**Chuan He**                                                          *he000233@umn.edu*
*Department of Computer Science and Engineering, University of Minnesota*

**Le Peng**                                                          *peng0347@umn.edu*
*Department of Computer Science and Engineering, University of Minnesota*

**Ju Sun**                                                          *jusun@umn.edu*
*Department of Computer Science and Engineering, University of Minnesota*

**Reviewed on OpenReview:** *https://openreview.net/forum?id=qItxVbWyfe*

## Abstract

In practice, many machine learning (ML) problems come with constraints, and their applied domains involve distributed sensitive data that cannot be shared with others, e.g., in healthcare. Collaborative learning in such practical scenarios entails federated learning (FL) for ML problems with constraints, or *FL with constraints* for short. Despite the extensive developments of FL techniques in recent years, these techniques only deal with unconstrained FL problems or FL problems with simple constraints that are amenable to easy projections. There is little work dealing with FL problems with general constraints. To fill this gap, we take the first step toward building an algorithmic framework for solving FL problems with general constraints. In particular, we propose a new FL algorithm for constrained ML problems based on the proximal augmented Lagrangian (AL) method. Assuming convex objective and convex constraints plus other mild conditions, we establish the worst-case complexity of the proposed algorithm. Our numerical experiments show the effectiveness of our algorithm in performing Neyman-Pearson classification and fairness-aware learning with nonconvex constraints, in an FL setting.

## 1 Introduction

Federated learning (FL) has emerged as a prominent distributed machine learning (ML) paradigm that respects data privacy by design and has found extensive applications in diverse domains (Kairouz et al., 2021). In FL, ML models are trained without centralized training data: local clients hold their local data and never directly share them with other clients or the central server. Given a global ML model to train, typical FL strategies consist of repeated local computation and central aggregation: in each round, each local client performs local computation of quantities of interest (e.g., local model parameters or derivatives) based on the local data, and then the central server collects and aggregates the local results and updates the parameters of the global ML model. Since the shared local results are usually highly nonlinear functions of local data, making reverse engineering of local data unlikely, data privacy is naturally protected.

### 1.1 Federated learning for constrained machine learning problems

However, existing FL techniques are developed almost exclusively for unconstrained ML problems or, at best, for ML problems with simple constraints that are amenable to easy projections, despite the growing list of ML problems with general constraints—where constraints typically encode prior knowledge and desired properties, e.g., robustness evaluation (Goodfellow et al., 2014), fairness-aware learning (Agarwal et al., 2018), learning with imbalanced data (Saito & Rehmsmeier, 2015), neural architecture search (Zoph et al.,

2018), topology optimization (Christensen & Klarbring, 2008), physics-informed machine learning (McClenny & Braga-Neto, 2020). Here, we sketch two quick examples.

**Neyman-Pearson classification, or optimizing the false-positive rate with a controlled false-negative rate** Conventional binary classification assumes equal importance in both classes, so predictive errors in both classes are counted equally. In numerous applications, such as medical diagnosis, misclassifying one class (i.e., the priority class) is much more costly than misclassifying the other. The Neyman-Pearson classification framework addresses this asymmetry in misclassification cost by explicitly controlling the error rate in the priority class while optimizing that in the other (Tong et al., 2016; Scott, 2007; Rigollet & Tong, 2011; Tong et al., 2018):

$$\min_{\theta} \frac{1}{n_0} \sum_{i=1}^{n_0} \varphi(f_\theta, z_{i,0}) \quad \text{s.t.} \quad \frac{1}{n_1} \sum_{i=1}^{n_1} \varphi(f_\theta, z_{i,1}) \leq r, \tag{1}$$

where $f_\theta$ is the trainable binary classifier parameterized by $\theta$, $\varphi$ is the loss function serving as a proxy to classification error, and $\{z_{i,0}\}_{i=1}^{n_0}$ and $\{z_{i,1}\}_{i=1}^{n_1}$ are the training data from class 0 and 1, respectively. The constraint imposes an upper bound on the error rate for class 1.

**Fairness-aware learning** Typical ML models are known to have biases toward the majority subgroups of the input space (Agarwal et al., 2018; Celis et al., 2019; Mehrabi et al., 2021). For example, a disease diagnostic model that is trained on a male-dominant dataset tends to predict much more accurately on the male subgroup than on the female subgroup. To counteract such potential model biases, a natural way is to enforce fairness constraints to ensure that the performance of the model on different subgroups is comparable (Agarwal et al., 2018; Celis et al., 2019; Mehrabi et al., 2021). A possible formulation for two-subgroup problems is

$$\min_{\theta} \frac{1}{n'} \sum_{i=1}^{n'} \varphi(f_\theta, z_i) \quad \text{s.t.} \quad -\delta \leq \frac{1}{|\mathcal{S}_0|} \sum_{i \in \mathcal{S}_0} \varphi(f_\theta, z_i) - \frac{1}{|\mathcal{S}_1|} \sum_{i \in \mathcal{S}_1} \varphi(f_\theta, z_i) \leq \delta, \tag{2}$$

where $f_\theta$ is the ML model parameterized by $\theta$, $\{z_i\}$ is the training set, and $\varphi$ is the proxy loss function, similar to the setup in Eq. (1). With $\mathcal{S}_0$ and $\mathcal{S}_1$ denoting the two subgroups of interest, the constraint imposes that the performance disparity of $f_\theta$ on $\mathcal{S}_0$ and $\mathcal{S}_1$ should not be larger than $\delta > 0$, which is usually set close to 0.

Both examples are particularly relevant to biomedical problems, where class imbalance and subgroup imbalance are prevalent. Moreover, there are strict regulations on the distribution and centralization of biomedical data for research, e.g., the famous Health Insurance Portability and Accountability Act (HIPAA) protection of patient privacy (OCR, 2023). Together, these underscore the importance of developing FL techniques for constrained ML problems, which is largely lacking: FL problems with only simple constraints that are amenable to easy projections have been considered in Yuan et al. (2021); Tran Dinh et al. (2021), and a small number of papers have tried to mitigate class imbalance (Shen et al., 2021) and improve model fairness (Du et al., 2021; Chu et al., 2021; Gálvez et al., 2021) through constrained optimization in FL settings. However, these developments are specialized to their particular use cases and lack computational guarantees for the feasibility and optimality of their solutions.

*In this paper, we take the first step toward a general and rigorous FL framework for general constrained ML problems.* Consider the constrained ML problems in Eqs. (1) and (2) in an FL setting with $n$ local clients, where the $i^{th}$ client holds local data $Z_i$ and so the whole training set is the union $Z_1 \cup \cdots \cup Z_n$. Since the objectives and constraints in Eqs. (1) and (2) are in the finite-sum form, both examples are special cases of the following finite-sum constrained optimization problem:

$$\min_{\theta} \left\{ \sum_{i=1}^{n} f_i(\theta; Z_i) + h(\theta) \right\} \quad \text{s.t.} \quad \underbrace{\sum_{i=1}^{n} \tilde{c}_i(\theta; Z_i) \leq 0.}_{\text{local data coupled}} \tag{3}$$

Note that inside the constraint, the local data are coupled which necessarily lead to communication between the local clients and the central server to allow the evaluation of the constraint function. To try to reduce such communication so that we can have more flexibility in algorithm design, we introduce decoupling variables $\{s_i\}$, leading to an equivalent formulation:

$$\min_{\theta, s_i} \left\{ \sum_{i=1}^{n} f_i(\theta; Z_i) + h(\theta) \right\} \quad \text{s.t.} \quad \underbrace{\sum_{i=1}^{n} s_i \leq 0}_{\text{no local data}}, \quad \underbrace{\tilde{c}_i(\theta; Z_i) \leq s_i, \ \ 1 \leq i \leq n}_{\text{local data decoupled}}, \tag{4}$$

which decouples the local data into $n$ local constraints.

In this paper, we consider the following setup for FL with global and local constraints, a strict generalization of Eq. (4):

$$\min_{w} \left\{ \sum_{i=1}^{n} f_i(w; Z_i) + h(w) \right\} \quad \text{s.t.} \quad \underbrace{c_0(w; Z_0) \leq 0}_{\text{global constraint}}, \quad \underbrace{c_i(w; Z_i) \leq 0, \ \ 1 \leq i \leq n}_{\text{local constraints}}. \tag{5}$$

In contrast to unconstrained FL or FL with simple constraints amenable to easy projections studied in existing literature, our focus lies on general convex constraints, where projections may or may not be easy to compute. Here, we assume $n$ local clients, each with a local objective $f_i(w; Z_i)$ and a set of local constraints $c_i(w; Z_i) \leq 0$ (i.e., scalar constraints are vectorized) for $i = 1, \ldots, n$. To stress the FL setting, we spell out the dependency of these local objectives and local constraints on the local data $Z_i$'s: *these $Z_i$'s are only accessible to their respective local clients and should never be shared with other local clients or the central server.* Henceforth, we omit $Z_i$'s in local objectives and constraints when no confusion arises. To allow flexibility in modeling, we also include the global constraint $c_0(w; Z_0) \leq 0$, with central data $Z_0$ that is only accessible by the central server. To facilitate the theoretical study of the algorithm that we develop, we further assume that all objective and scalar constraint functions are convex, but we also verify the applicability of our FL algorithm to classification problems with nonconvex fairness constraints in Section 5.2. To summarize, our standing assumptions on top of Eq. (5) include

**Assumption 0.** *We make the following assumption throughout this paper:*

*(a) The objective functions $f_i : \mathbb{R}^d \to \mathbb{R}$, $1 \leq i \leq n$, and the $m_i$ scalar constraint functions inside $c_i : \mathbb{R}^d \to \mathbb{R}^{m_i}$, $0 \leq i \leq n$, are convex and continuously differentiable, and $h : \mathbb{R}^d \to (-\infty, \infty]$ is a simple closed convex function.*

*(b) For each $1 \leq i \leq n$, only the local objective $f_i$ and local constraint $c_i$ have access to the local data $Z_i$, which are never shared with other local clients and the central server. Only the central server has access to the global data $Z_0$,*

## 1.2 Our contributions

This paper tackles Eq. (5) by adopting the sequential penalization approach, which involves solving a sequence of unconstrained subproblems that combine the objective function with penalization of constraint violations. In particular, we propose an FL algorithm based on the proximal augmented Lagrangian (AL) method developed in Lu & Zhou (2023). In each iteration, the unconstrained subproblem is solved by an inexact solver based on the alternating direction method of multipliers (ADMM) in a federated manner. We study the worst-case complexity of the proposed algorithm assuming *locally* Lipschitz continuous gradients. Our main contributions are highlighted below.

- We propose an FL algorithm (Algorithm 1) for solving Eq. (5) based on the proximal AL method. To the best of our knowledge, the proposed algorithm is the first in solving general constrained ML problems in an FL setting. Assuming *locally* Lipschitz continuous gradients and other mild conditions, we establish its worst-case complexity to find an approximate optimal solution of Eq. (5). The complexity results are entirely new in the literature.
- We propose an ADMM-based inexact solver in Algorithm 2 to solve the unconstrained subproblems arising in Algorithm 1. We equip this inexact solver with a newly introduced verifiable termination criterion and

establish its global linear convergence for solving the subproblems of Algorithm 1; these subproblems are strongly convex and have locally Lipschitz continuous gradients.

- We perform numerical experiments to compare our proposed FL algorithm (Algorithm 1) with the centralized proximal AL method (Algorithm 3) on binary Neyman-Pearson classification and classification with nonconvex fairness constraints using real-world datasets (Section 5). Our numerical results demonstrate that our FL algorithm can achieve solution quality comparable to that of the centralized proximal AL method.

### 1.3 Related work

**FL algorithms for unconstrained optimization**  FL has emerged as a cornerstone for privacy-preserved learning since Google's seminal work (McMahan et al., 2017), and has found applications in numerous domains where the protection of data privacy precludes centralized learning, including healthcare (Rieke et al., 2020; Peng et al., 2023a;b), finance (Long et al., 2020), Internet of things (Mills et al., 2019), and transportation Liu et al. (2020). FedAvg (McMahan et al., 2017) is the first and also the most popular FL algorithm to date. After FedAvg, numerous FL algorithms have been proposed to improve performance and address practical issues, such as data heterogeneity (Karimireddy et al., 2020; Li et al., 2021c; Zhang et al., 2021), system heterogeneity (Li et al., 2020; Wang et al., 2020; Gong et al., 2022), fairness (Li et al., 2021b), communication efficiency (Sattler et al., 2019; Konečnỳ et al., 2016; Mishchenko et al., 2022), convergence (Pathak & Wainwright, 2020), handling simple constraints (Yuan et al., 2021; Tran Dinh et al., 2021), incentives (Travadi et al., 2023), and hyperparameter tuning (Yao et al., 2024). Since our FL algorithm relies on applying an inexact ADMM (Algorithm 2) to solve subproblems, it is also worth mentioning that ADMM-based algorithms have been proposed to handle FL problems (Zhou & Li, 2023; Gong et al., 2022; Zhang et al., 2021) and optimization problems with many constraints in a distributed manner (Giesen & Laue, 2019). More FL algorithms and their applications can be found in the survey (Li et al., 2021a). Despite the intensive research on FL, existing algorithms focus primarily on unconstrained ML problems, versus constrained ML problems considered in this paper.

**Centralized algorithms for constrained optimization**  Recent decades have seen fruitful algorithm developments for centralized constrained optimization in numerical optimization. In particular, there has been a rich literature on AL methods for solving convex constrained optimization problems (Aybat & Iyengar, 2013; Necoara et al., 2019; Patrascu et al., 2017; Xu, 2021; Lan & Monteiro, 2016; Lu & Zhou, 2023; Lu & Mei, 2023). In addition, variants of AL methods have been developed to solve nonconvex constrained optimization problems (Hong et al., 2017; Grapiglia & Yuan, 2021; Birgin & Martínez, 2020; Kong et al., 2023; Li et al., 2021d; He et al., 2023a;b; Lu, 2022). Besides AL methods and their variants, sequential quadratic programming methods (Boggs & Tolle, 1995; Curtis & Overton, 2012), trust-region methods (Byrd et al., 1987; Powell & Yuan, 1991), interior point methods (Wächter & Biegler, 2006), and extra-point methods Huang et al. (2022) have been proposed to solve centralized constrained optimization problems.

**Distributed algorithms for constrained optimization**  Developing distributed algorithms for constrained optimization has started relatively recently. To handle simple local constraints in distributed optimization, Nedic et al. (2010); Lin et al. (2016); Wang et al. (2017) study distributed projected subgradient methods. For complicated conic local constraints, Aybat & Hamedani (2016; 2019) develop distributed primal-dual algorithms. For distributed optimization with global and local constraints, Zhu & Martínez (2011); Yuan et al. (2011) develop primal-dual projected subgradient algorithms. For an overview of distributed constrained optimization, see Yang et al. (2019). Notice that FL is a special distributed optimization/learning framework that protects data privacy by prohibiting the transfer of raw data from one client to another or to a central server. These distributed algorithms for constrained optimization do not violate the FL restriction and hence can be considered as FL algorithms, but they can only handle problems with simple global or local constraints that are amenable to easy projection. Therefore, they cannot be applied directly to our setup Eq. (5) with general global and local constraints.

**FL algorithms for constrained ML applications**  A small number of papers have developed FL algorithms for particular constrained ML applications, such as learning with class imbalance and fairness-aware

ML. For example, Shen et al. (2021); Chu et al. (2021) propose FL algorithms to address class imbalance and subgroup imbalance, respectively, by optimizing the Lagrangian function. Du et al. (2021) applies quadratic penalty method to deal with the constraint in fairness-aware ML. In addition, Gálvez et al. (2021) proposes an FL algorithm to tackle fairness-aware ML based on optimizing the AL function. However, these developments are tailored to specific applications and lack rigorous computational guarantees regarding the feasibility and optimality of their solutions. In contrast, this paper focuses on developing algorithms with theoretical guarantees for FL with convex global and local constraints. To the best of our knowledge, this work provides the first general FL framework for constrained ML problems.

## 2  Notation and preliminaries

Throughout this paper, we let $\mathbb{R}^d$ and $\mathbb{R}_+^d$ denote the $d$-dimensional Euclidean space and its nonnegative orthant, respectively. We use $\langle \cdot, \cdot \rangle$ to denote the standard inner product, $\| \cdot \|$ to denote the Euclidean norm of a vector or the spectral norm of a matrix, and $\| \cdot \|_\infty$ to denote the $\ell_\infty$-norm of a vector. For any vector $v \in \mathbb{R}^d$, $[v]_+ \in \mathbb{R}^d$ is its nonnegative part (i.e., with all negative values set to zero). We adopt the standard big-O notation $\mathcal{O}(\cdot)$ to present complexity results; $\widetilde{\mathcal{O}}(\cdot)$ represents $\mathcal{O}(\cdot)$ with logarithmic terms omitted.

Given a closed convex function $h : \mathbb{R}^d \to (-\infty, \infty]$, $\partial h$ and $\mathrm{dom}(h)$ denote the subdifferential and domain of $h$, respectively. The proximal operator associated with $h$ is denoted by $\mathrm{prox}_h$, that is, $\mathrm{prox}_h(u) = \arg\min_w \{ \|w - u\|^2/2 + h(w) \}$ for all $u \in \mathbb{R}^d$. Given a continuously differentiable mapping $\phi : \mathbb{R}^d \to \mathbb{R}^p$, we write the transpose of its Jacobian as $\nabla\phi(w) = [\nabla\phi_1(w) \cdots \nabla\phi_p(w)] \in \mathbb{R}^{d \times p}$. We say that $\nabla\phi$ is $L$-Lipschitz continuous on a set $\Omega$ for some $L > 0$ if $\|\nabla\phi(u) - \nabla\phi(v)\| \le L\|u - v\|$ for all $u, v \in \Omega$. In addition, we say that $\nabla\phi$ is locally Lipschitz continuous on $\Omega$ if for any $w \in \Omega$, there exist $L_w > 0$ and an open set $\mathcal{U}_w$ containing $w$ such that $\nabla\phi$ is $L_w$-Lipschitz continuous on $\mathcal{U}_w$.

Given a nonempty closed convex set $\mathcal{C} \subseteq \mathbb{R}^d$ and any point $u \in \mathbb{R}^d$, $\mathrm{dist}(u, \mathcal{C})$ and $\mathrm{dist}_\infty(u, \mathcal{C})$ stand for the Euclidean distance and the Chebyshev distance from $u$ to $\mathcal{C}$, respectively. That is, $\mathrm{dist}(u, \mathcal{C}) = \min_{v \in \mathcal{C}} \|u - v\|$ and $\mathrm{dist}_\infty(u, \mathcal{C}) = \min_{v \in \mathcal{C}} \|u - v\|_\infty$. The normal cone of $\mathcal{C}$ at $u \in \mathcal{C}$ is denoted by $\mathcal{N}_\mathcal{C}(u)$. The Minkowski sum of two sets $\mathcal{B}$ and $\mathcal{C}$ is defined as $\mathcal{B} + \mathcal{C} := \{b + c : b \in \mathcal{B}, c \in \mathcal{C}\}$.

For ease of presentation, we let $m := \sum_{i=0}^n m_i$ and adopt the following notations throughout this paper:

$$f(w) = \sum_{i=1}^n f_i(w), \qquad c(w) = \begin{bmatrix} c_0(w) \\ \vdots \\ c_n(w) \end{bmatrix} \in \mathbb{R}^m, \qquad \mu = \begin{bmatrix} \mu_0 \\ \vdots \\ \mu_n \end{bmatrix} \in \mathbb{R}^m. \tag{6}$$

**Assumption 1.** *Throughout this paper, we assume that the strong duality holds for Eq. (5) and its dual problem*

$$\sup_{\mu \ge 0} \inf_w \{f(w) + h(w) + \langle \mu, c(w) \rangle\}. \tag{7}$$

*That is, both problems have optimal solutions and, moreover, their optimal values coincide.*

Under Assumption 1, it is known that $(w, \mu) \in \mathrm{dom}(h) \times \mathbb{R}_+^m$ is a pair of optimal solutions of Eq. (5) and Eq. (7) if and only if it satisfies (see, e.g., Lu & Zhou (2023))

$$0 \in \begin{pmatrix} \nabla f(w) + \partial h(w) + \nabla c(w)\mu \\ c(w) - \mathcal{N}_{\mathbb{R}_+^m}(\mu) \end{pmatrix}. \tag{8}$$

In general, it is hard to find an exact optimal solution of Eq. (5) and Eq. (7). Thus, we are instead interested in seeking an approximate optimal solution of Eq. (5) and Eq. (7) defined as follows.

**Definition 1.** *Given any $\epsilon_1, \epsilon_2 > 0$, we say $(w, \mu) \in \mathrm{dom}(h) \times \mathbb{R}_+^m$ is an $(\epsilon_1, \epsilon_2)$-optimal solution of Eq. (5) and Eq. (7) if $\mathrm{dist}_\infty(0, \nabla f(w) + \partial h(w) + \nabla c(w)\mu) \le \epsilon_1$ and $\mathrm{dist}_\infty(c(w), \mathcal{N}_{\mathbb{R}_+^m}(\mu)) \le \epsilon_2$.*[1]

---

[1] For unconstrained convex problems with differentiable objective $\min_w f(w)$, a natural measure of convergence is $\|\nabla f(w)\|$, i.e., the distance between 0 and $\nabla f(w)$, as the optimality condition is $\nabla f(w) = 0$. If the objective is nondifferentiable, we need to use the notation of subdifferential, $\partial f(w)$, which is a set for each $w$ in general. In this case, the optimality condition reads $0 \in \partial f(w)$, and the measure of convergence is the distance between 0 and the subdifferent set $\mathrm{dist}(0, \partial f(w)) := \min_{u \in \partial f(w)} \|u\|$.

Here, the two different tolerances $\epsilon_1, \epsilon_2$ are used for measuring stationarity and feasibility violation, respectively. This definition is consistent with the $\epsilon$-KKT solution considered in Lu & Zhou (2023) except that Definition 1 uses the Chebyshev distance rather than the Euclidean distance.

## 3  A proximal AL based FL algorithm for solving Eq. (5)

In this section, we propose an FL algorithm for solving Eq. (5) based on the proximal AL method. Specifically, we describe this algorithm in Section 3.1, and then analyze its complexity results in Section 3.2.

**Assumption 2.** *Throughout this section, we assume that*

*(a) The proximal operator for $h$ can be exactly evaluated.*

*(b) The gradients $\nabla f_i$, $1 \le i \le n$, and the transposed Jocobians $\nabla c_i$, $0 \le i \le n$, are locally Lipschitz continuous on $\mathbb{R}^d$.*

Assumption 2(b) clearly holds if all $\nabla f_i$'s and $\nabla c$'s are globally Lipschitz continuous on $\mathbb{R}^d$, but this assumption holds for a broad class of problems without global Lipschitz continuity on $\nabla f_i$'s and $\nabla c_i$'s. For example, the quadratic penalty function of $c(w) \le 0$, namely $\|[c(w)]_+\|^2$, only has a locally Lipschitz continuous gradient even if $\nabla c$ is globally Lipschitz continuous on $\mathbb{R}^d$ (see Remark 4.1). In addition, the gradient of a convex high-degree polynomial, such as $\|w\|^4$ with $w \in \mathbb{R}^d$, is locally Lipschitz continuous but not globally Lipschitz continuous on $\mathbb{R}^d$.

### 3.1  Algorithm description

In this subsection, we describe a proximal AL-based FL algorithm (Algorithm 1) for finding an $(\epsilon_1, \epsilon_2)$-optimal solution of Eq. (5) for prescribed $\epsilon_1, \epsilon_2 \in (0, 1)$. This algorithm follows a framework similar to a centralized proximal AL method described in Appendix D; see Section 11.K in Rockafellar & Wets (2009) or Lu & Zhou (2023) for more details of proximal AL. At each iteration, it applies an inexact ADMM solver (Algorithm 2) to find an approximate solution $w^{k+1}$ to the proximal AL subproblem associated with Eq. (5):

$$\min_w \left\{ \ell_k(w) := \underbrace{\sum_{i=1}^n f_i(w) + h(w) + \frac{1}{2\beta} \sum_{i=0}^n \left( \|[\mu_i^k + \beta c_i(w)]_+\|^2 - \|\mu_i^k\|^2 \right)}_{\text{augmented Lagrangian function}} + \underbrace{\frac{1}{2\beta} \|w - w^k\|^2}_{\text{proximal term}} \right\}. \tag{9}$$

Then, the multiplier estimates are updated according to the classical scheme:

$$\mu_i^{k+1} = [\mu_i^k + \beta c_i(w^{k+1})]_+, \quad 0 \le i \le n.$$

Notice that the subproblem in Eq. (9) can be rewritten as

$$\min_w \left\{ \ell_k(w) := \sum_{i=0}^n P_{i,k}(w) + h(w) \right\}, \tag{12}$$

where $P_{i,k}$, $0 \le i \le n$, are defined as

$$P_{0,k}(w) := \frac{1}{2\beta} \left( \|[\mu_0^k + \beta c_0(w)]_+\|^2 - \|\mu_0^k\|^2 \right) + \frac{1}{2(n+1)\beta} \|w - w^k\|^2, \tag{13}$$

$$P_{i,k}(w) := f_i(w) + \frac{1}{2\beta} \left( \|[\mu_i^k + \beta c_i(w)]_+\|^2 - \|\mu_i^k\|^2 \right) + \frac{1}{2(n+1)\beta} \|w - w^k\|^2, \quad \forall 1 \le i \le n. \tag{14}$$

When Algorithm 2 (see Section 4) is applied to solve Eq. (12), the local merit function $P_{i,k}$, constructed from the local objective $f_i$ and local constraint $c_i$, is handled by the respective local client $i$, while the merit function $P_{0,k}$ is handled by the central server. We observe that Algorithm 1 with the subproblem in Eq. (12) solved by Algorithm 2 meets the basic FL requirement: since local objective $f_i$'s and local constraint $c_i$'s are handled by their respective local clients and the central server only performs aggregation and handles the global constraint $c_0$, no raw data are shared between local clients and the central server, i.e., Assumption 0(b) is obeyed.

---

**Algorithm 1** A proximal AL based FL algorithm for solving Eq. (5)

---

**Input**: tolerances $\epsilon_1, \epsilon_2 \in (0, 1)$, $w^0 \in \text{dom}(h)$, $\mu_i^0 \geq 0$ for $0 \leq i \leq n$, $\bar{s} > 0$, and $\beta > 0$.

1: **for** $k = 0, 1, 2, \dots$ **do**
2:     Set $\tau_k = \bar{s}/(k+1)^2$.
3:     Call Algorithm 2 (see Section 4 below) with $(\tau, \tilde{w}^0) = (\tau_k, w^k)$ to find an approximate solution $w^{k+1}$
4:     to Eq. (12) in a federated manner such that

$$\text{dist}_\infty(0, \partial \ell_k(w^{k+1})) \leq \tau_k. \tag{10}$$

5:     **Server update:** The central server updates $\mu_0^{k+1} = [\mu_0^k + \beta c_0(w^{k+1})]_+$.
6:     **Communication (broadcast):** Each local client $i$, $1 \leq i \leq n$, receives $w^{k+1}$ from the central server.
7:     **Client update (local):** Each local client $i$, $1 \leq i \leq n$, updates $\mu_i^{k+1} = [\mu_i^k + \beta c_i(w^{k+1})]_+$.
8:     **Communication:** Each local client $i$, $1 \leq i \leq n$, sends $\|\mu_i^{k+1} - \mu_i^k\|_\infty$ to the central server.
9:     **Termination (server side):** Output $(w^{k+1}, \mu^{k+1})$ and terminate the algorithm if

$$\|w^{k+1} - w^k\|_\infty + \beta \tau_k \leq \beta \epsilon_1, \qquad \max_{0 \leq i \leq n} \{\|\mu_i^{k+1} - \mu_i^k\|_\infty\} \leq \beta \epsilon_2. \tag{11}$$

10: **end for**

---

**Remark 3.1.** *We now make the following remarks on Algorithm 1.*

*(a) For hyperparameters of Algorithm 1,*
  - $\epsilon_1, \epsilon_2 \in (0, 1)$ *only depend on the numerical accuracy that the user aims to achieve;*
  - *the initial iterates $w^0$ and $\mu_i^0$, $1 \leq i \leq n$, are usually randomly generated or set as a constant vector;*
  - $\bar{s} > 0$ *controls the tolerance sequence $\{\tau_k\}_{k \geq 0}$ for the subproblems in Algorithm 1. These finite, non-zero tolerances allow us to solve the subproblems inexactly but can still guarantee convergence, hence saving computational costs. In particular, setting $\{\tau_k\}_{k \geq 0}$ to diminish rapidly towards zero on the order of $\mathcal{O}(1/k^2)$ can guarantee convergence of Algorithm 1. In practice, $\bar{s}$ only needs to be set as $\mathcal{O}(1)$.*

*(b) Compared to the centralized proximal AL developed in Lu & Zhou (2023), we have made the following major changes to arrive at Algorithm 1.*
  - *add communication steps to allow dual updates in an FL manner;*
  - *to solve the subproblem, we cannot directly apply the accelerated gradient method (AGM) as in Lu & Zhou (2023). It is possible to develop an FL version of AGM by eagerly aggregating gradients from local clients, but that induces heavy communication between clients and the central server. To address this, we first reformulate the subproblem as a finite-sum problem and then propose an inexact ADMM solver to solve it. The inexact ADMM solver allows multiple steps of local updates before aggregation of model weights at the central server, hence it is communication friendly. We also propose a new stopping criterion for the inexact ADMM (Algorithm 2). Detailed explanations can be found in Remark 4.2.*

For ease of later reference, we refer to the update from $w^k$ to $w^{k+1}$ as one outer iteration of Algorithm 1, and call one iteration of Algorithm 2 for solving Eq. (9) one inner iteration of Algorithm 1. In the rest of this section, we study the following measures of complexity for Algorithm 1.

- *Outer iteration complexity*, which measures the number of outer iterations of Algorithm 1 (*one outer iteration* refers to one execution from Line 2 to Line 9 in Algorithm 1);
- *Total inner iteration complexity*, which measures the total number of iterations of Algorithm 2 that are performed in Algorithm 1 (*one inner iteration* refers to one execution from Line 3 to Line 10 in Algorithm 2).

The following theorem concerns the output of Algorithm 1, whose proof is deferred to Appendix A.1.

**Theorem 3.1** (**output of Algorithm 1**)**.** *If Algorithm 1 successfully terminates, its output $(w^{k+1}, \mu^{k+1})$ is an $(\epsilon_1, \epsilon_2)$-optimal solution of Eq. (5).*

## 3.2 Complexity analysis

In this subsection, we establish the outer and total inner iteration complexity for Algorithm 1. To proceed, we let $(w^*, \mu^*)$ be any pair of optimal solutions of Eq. (5) and Eq. (7). First, we establish a lemma to show that all iterates generated by Algorithm 1 are bounded. Its proof can be found in Appendix A.2.

**Lemma 3.1 (bounded iterates of Algorithm 1).** *Suppose that Assumptions 0 to 2 hold. Let $\{w^k\}_{k \geq 0}$ be all the iterates generated by Algorithm 1. Then we have $w^k \in \mathcal{Q}_1$ for all $k \geq 0$, where*

$$\mathcal{Q}_1 := \{w \in \mathbb{R}^d : \|w - w^*\| \leq r_0 + 2\sqrt{n}\bar{s}\beta\} \quad \text{with } r_0 := \|(w^0, \mu^0) - (w^*, \mu^*)\|, \tag{15}$$

*and $w^0$, $\mu^0$, $\bar{s}$, and $\beta$ are inputs of Algorithm 1.*

This boundedness result allows us to utilize the Lipschitz continuity on a bounded set to establish the convergence rate for Algorithm 1. The following theorem states the worst-case complexity results of Algorithm 1, whose proof is relegated to Appendix A.3.

**Theorem 3.2 (complexity results of Algorithm 1).** *Suppose that Assumptions 0 to 2 hold. Then,*

*(a) the number of outer iteration of Algorithm 1 is at most $\mathcal{O}(\max\{\epsilon_1^{-2}, \epsilon_2^{-2}\})$; and*

*(b) the total number of inner iterations of Algorithm 1 is at most $\widetilde{\mathcal{O}}(\max\{\epsilon_1^{-2}, \epsilon_2^{-2}\})$.*

**Remark 3.2.** *(a) To the best of our knowledge, Theorem 3.2 provides the first worst-case complexity results for finding an approximate optimal solution of Eq. (5) in an FL framework; (b) The number of outer and inner iterations of Algorithm 1 with detailed dependencies on the algorithm hyperparameters can be found in Eqs. (48) and (84) in the proofs, respectively.*

## 3.3 Communication overheads

In the outer loop of Algorithm 1, a single communication round occurs after solving a proximal AL subproblem. During this round, the central server sends the current weights $w^{k+1}$ to all local clients, and each client sends back the maximum change in their respective multipliers, measured by $\|\mu_i^{k+1} - \mu_i^k\|_\infty$, to the central server. The communication overheads of the inner solver Algorithm 2 are discussed in Section 4.3. The communication complexity of Algorithm 1 is $\widetilde{\mathcal{O}}(\max\{\epsilon_1^{-2}, \epsilon_2^{-2}\})$.

# 4 An inexact ADMM for FL

In this section, we propose an inexact ADMM-based FL algorithm to solve the subproblem in Eq. (12) (the same as Eq. (9)) for Algorithm 1. Before proceeding, we show that $\nabla P_{i,k}$, $0 \leq i \leq n$, are locally Lipschitz continuous on $\mathbb{R}^d$, whose proof is deferred to Appendix B.1.

**Lemma 4.1 (local Lipschitz continuity of $\nabla P_{i,k}$).** *Suppose that Assumptions 0 to 2 hold. Then the gradients $\nabla P_{i,k}$, $0 \leq i \leq n$, are locally Lipschitz continuous on $\mathbb{R}^d$.*

**Remark 4.1.** *It is worth noting that $\nabla P_{i,k}$, $0 \leq i \leq n$, are typically not globally Lipschitz continuous on $\mathbb{R}^d$ even if $\nabla f_i$, $1 \leq i \leq n$, and $\nabla c_i$, $0 \leq i \leq n$, are globally Lipschitz continuous on $\mathbb{R}^d$. For example, consider $c_0(w) = \|w\|^2 - 1$. By Eq. (13), one has that*

$$\nabla P_{0,k}(w) = 2[\mu_0^k + \beta(\|w\|^2 - 1)]_+ w + \frac{1}{(n+1)\beta}(w - w^k).$$

*In this case, it is not hard to verify that $\nabla c_0$ is globally Lipschitz continuous on $\mathbb{R}^d$, but $\nabla P_{0,k}$ is not. Thus, analyzing the complexity results for solving the subproblems in Eq. (12) using local Lipschitz conditions of $\nabla P_{i,k}$, $0 \leq i \leq n$, is reasonable.*

Moreover, it is easy to see that $P_{i,k}$ are strongly convex with the modulus $1/[(n+1)\beta]$ for all $0 \leq i \leq n$ and all $k \geq 0$.

Since both the local Lipschitz and the strong convexity (including its modulus) properties hold for all $k \geq 0$, and we need to solve the subproblem of the same form each $k$, below we drop $k$ and focus on solving the

following model problem in an FL manner:

$$\min_w \left\{ \ell(w) := \sum_{i=0}^n P_i(w; Z_i) + h(w) \right\}, \tag{16}$$

where the data $Z_i$'s are only accessible to their corresponding local/global functions $P_i$'s, necessitating FL. We will drop $Z_i$'s henceforth for simplicity. The model problem in Eq. (16) satisfies:

1. The functions $P_i$, $0 \le i \le n$, are continuously differentiable, and moreover, $\nabla P_i$, $0 \le i \le n$, are locally Lipschitz continuous on $\mathbb{R}^d$;
2. The functions $P_i$, $0 \le i \le n$, are strongly convex with a modulus $\sigma > 0$ on $\mathbb{R}^d$, that is,

$$\langle \nabla P_i(u) - \nabla P_i(v), u - v \rangle \ge \sigma \|u - v\|^2, \quad \forall u, v \in \mathbb{R}^d, \ 0 \le i \le n. \tag{17}$$

### 4.1 Algorithm description

---

**Algorithm 2** An inexact ADMM based FL algorithm for solving Eq. (16)

---

**Input**: tolerance $\tau \in (0, 1]$, $q \in (0, 1)$, $\tilde{w}^0 \in \text{dom}(h)$, and $\rho_i > 0$ for $1 \le i \le n$;

1: Set $w^0 = \tilde{w}^0$, and $(u_i^0, \lambda_i^0, \tilde{u}_i^0) = (\tilde{w}^0, -\nabla P_i(\tilde{w}^0), \tilde{w}^0 - \nabla P_i(\tilde{w}^0)/\rho_i)$ for $1 \le i \le n$.

2: **for** $t = 0, 1, 2, \dots$ **do**

3:     Set $\varepsilon_{t+1} = q^t$;

4:     **Server update:** The central server finds an approximate solution $w^{t+1}$ to

$$\min_w \left\{ \varphi_{0,t}(w) := P_0(w) + h(w) + \sum_{i=1}^n \left[ \frac{\rho_i}{2} \|\tilde{u}_i^t - w\|^2 \right] \right\} \tag{18}$$

5:     such that $\text{dist}_\infty(0, \partial \varphi_{0,t}(w^{t+1})) \le \varepsilon_{t+1}$.

6:     **Communication (broadcast):** Each local client $i$, $1 \le i \le n$, receives $w^{t+1}$ from the server.

7:     **Client update (local):** Each local client $i$, $1 \le i \le n$, finds an approximate solution $u_i^{t+1}$ to

$$\min_{u_i} \left\{ \varphi_{i,t}(u_i) := P_i(u_i) + \langle \lambda_i^t, u_i - w^{t+1} \rangle + \frac{\rho_i}{2} \|u_i - w^{t+1}\|^2 \right\} \tag{19}$$

8:     such that $\|\nabla \varphi_{i,t}(u_i^{t+1})\|_\infty \le \varepsilon_{t+1}$, and then updates

$$\lambda_i^{t+1} = \lambda_i^t + \rho_i(u_i^{t+1} - w^{t+1}), \tag{20}$$

$$\tilde{u}_i^{t+1} = u_i^{t+1} + \lambda_i^{t+1}/\rho_i, \tag{21}$$

$$\tilde{\varepsilon}_{i,t+1} = \|\nabla \varphi_{i,t}(w^{t+1}) - \rho_i(w^{t+1} - u_i^t)\|_\infty. \tag{22}$$

9:     **Communication:** Each local client $i$, $1 \le i \le n$, sends $(\tilde{u}_i^{t+1}, \tilde{\varepsilon}_{i,t+1})$ back to the central server.

10:    **Termination (server side):** Output $w^{t+1}$ and terminate this algorithm if

$$\varepsilon_{t+1} + \sum_{i=1}^n \tilde{\varepsilon}_{i,t+1} \le \tau. \tag{23}$$

11: **end for**

---

In this subsection, we propose an inexact ADMM-based FL algorithm (Algorithm 2) for solving Eq. (16). To make each participating client $i$ handle their local objective $P_i$ independently (see Section 3.1), we introduce decoupling variables $u_i$'s and obtain the following equivalent consensus reformulation for Eq. (16):

$$\min_{w,u_i} \left\{ \sum_{i=1}^n P_i(u_i) + P_0(w) + h(w) \right\} \quad \text{s.t.} \quad u_i = w, \quad 1 \le i \le n, \tag{24}$$

which allows each local client $i$ to handle the local variable $u_i$ and the local objective function $P_i$ while imposing consensus constraints that force clients' local parameters $u_i$ equal to the global parameter $w$. This reformulation enables the applicability of an inexact ADMM that solves Eq. (24) in a federated manner. At each iteration, an ADMM solver optimizes the AL function associated with Eq. (24):

$$\mathcal{L}_P(w, u, \lambda) := \sum_{i=1}^{n} \left[ P_i(u_i) + \langle \lambda_i, u_i - w \rangle + \frac{\rho_i}{2} \|u_i - w\|^2 \right] + P_0(w) + h(w) \tag{25}$$

with respect to the variables $w$, $u$, and $\lambda$ alternately, where $u = [u_1^T, \ldots, u_n^T]^T$ and $[\lambda_1^T, \ldots, \lambda_n^T]^T$ collect all the local parameters and the multipliers associated with the consensus constraints, respectively. Specifically, in iteration $t$, one performs

$$w^{t+1} \approx \arg\min_{w} \mathcal{L}_P(w, u^t, \lambda^t), \tag{26}$$

$$u^{t+1} \approx \arg\min_{u} \mathcal{L}_P(w^{t+1}, u, \lambda^t), \tag{27}$$

$$\lambda_i^{t+1} = \lambda_i^t + \rho_i(u_i^{t+1} - w^{t+1}), \quad \forall 1 \le i \le n. \tag{28}$$

By the definition of $\mathcal{L}_P$ in Eq. (25), one can verify that the step in Eq. (26) is equivalent to Eq. (18), and also the step in Eq. (27) can be computed in parallel, which corresponds to Eq. (19). Therefore, the ADMM updates naturally suit the FL framework, as the separable structure in Eq. (25) over the pairs $\{(u_i, \lambda_i)\}$ enables the local update of $(u_i, \lambda_i)$ at each client $i$ while $w$ is updated by the central server.

Since the subproblems in Eq. (18) and Eq. (19) are strongly convex, their approximate solutions $w^{t+1}$ and $u_i^{t+1}$, $1 \le i \le n$, can be found using a gradient-based algorithm with a global linear convergence rate (Nesterov et al., 2018). Furthermore, the value $\tilde{\varepsilon}_{i,t+1}$ in Eq. (22) serves as a measure of local optimality and consensus for client $i$. By summing up $\tilde{\varepsilon}_{i,t+1}$ for $1 \le i \le n$ and including $\varepsilon_{t+1}$, one can obtain a stationarity measure for the current iterate (see (Eq. (23))), as presented in the following theorem. Its proof can be found in Appendix B.2.

**Remark 4.2.** *We now make the following remarks on Algorithm 2.*

*(a) On hyperparameters of Algorithm 2,*
- *$(\tau, \tilde{w}^0)$ is specified as $(\tau_k, w^k)$ at the $k$th iteration of Algorithm 1.*
- *From Eq. (18), $\rho_i$, $1 \le i \le n$ can be viewed as weighting parameters for aggregation. Therefore, it is natural to set $\rho_i = am_i$ for $1 \le i \le n$, where $m_i$ is the number of samples in client $i$ and $a$ is a global constant. We follow this rule when setting $\rho_i$'s.*
- *$q \in (0, 1)$ determines the tolerance sequence $\{\varepsilon_{t+1}\}_{t \ge 0}$ for the subproblems in Eq. (18). These tolerances in solving subproblems reduce computational costs. Setting $\{\varepsilon_{t+1}\}_{t \ge 0}$ to rapidly diminish toward zero rapidly at a geometric rate ensures the convergence of Algorithm 1. In practice, we suggest setting $q$ as $\mathcal{O}(1)$.*

*(b) The main innovations we have here compared to the existing literature on ADMM-based FL algorithms (e.g, Zhou & Li (2023); Gong et al. (2022); Zhang et al. (2021)) include:*
- *We establish the complexity results of an inexact ADMM-based FL algorithm under local Lipschitz conditions, vs. global Lipschitz conditions in other work. Our complexity results can be found in Theorem 4.2.*
- *We propose a novel and rigorous stopping criterion (Eq. (23)) that is easily verifiable, communication-light, and compatible with the outer iterations (as our inexact ADMM FL algorithm serves as a subproblem solver in our overall algorithm framework).*

**Theorem 4.1 (output of Algorithm 2).** *If Algorithm 2 terminates at some iteration $T \ge 0$, then its output $w^{T+1}$ satisfies $\mathrm{dist}_\infty(0, \partial\ell(w^{T+1})) \le \tau$.*

Theorem 4.1 states that Algorithm 2 outputs a point that approximately satisfies the first-order optimality condition of Eq. (5). In addition, it follows from Theorem 4.1 that Algorithm 2 with $(\tau, \tilde{w}^0) = (\tau_k, w^k)$ finds an approximate solution $w^{k+1}$ to Eq. (12) such that Eq. (10) holds.

## 4.2 Complexity analysis

In this subsection, we establish the iteration complexity for the inexact ADMM, namely, Algorithm 2. Recall from Eq. (17) that Eq. (16) is strongly convex and thus has a unique optimal solution. We refer to this optimal solution of Eq. (16) as $\tilde{w}^*$ throughout this section. The following lemma shows that all the iterates generated by Algorithm 2 lie in a compact set. Its proof can be found in Appendix B.3.

**Lemma 4.2** (**bounded iterates of Algorithm 2**). *Suppose that Assumptions 0 to 2 hold and let $\{u_i^{t+1}\}_{1 \le i \le n, t \ge 0}$ and $\{w^{t+1}\}_{t \ge 0}$ be all the iterates generated by Algorithm 2. Then it holds that all these iterates stay in a compact set $\mathcal{Q}$, where*

$$\mathcal{Q} := \left\{ v : \|v - \tilde{w}^*\|^2 \le \frac{n+1}{\sigma^2(1-q^2)} + \frac{1}{\sigma} \sum_{i=1}^{n} \left( \rho_i \|\tilde{w}^* - \tilde{w}^0\|^2 + \frac{1}{\rho_i} \|\nabla P_i(\tilde{w}^*) - \nabla P_i(\tilde{w}^0)\|^2 \right) \right\}. \tag{29}$$

The iteration complexity of Algorithm 2 is established in the following theorem, whose proof is relegated to Appendix B.4.

**Theorem 4.2** (**iteration complexity of Algorithm 2**). *Suppose that Assumptions 0 to 2 hold. Then Algorithm 2 terminates in at most $\mathcal{O}(|\log \tau|)$ iterations.*

**Remark 4.3.** *We now make the following remarks on the complexity results in Theorem 4.2.*

*(a) Algorithm 2 enjoys a global linear convergence rate when solving the problem in Eq. (16). The result generalizes classical convergence results for ADMM in the literature, which typically require a strongly convex objective with globally Lipschitz continuous gradient (e.g., see Lin et al. (2015)). In contrast, our result is the first to establish a global linear convergence of an inexact ADMM assuming a strongly convex objective with only a locally Lipschitz continuous gradient.*

*(b) The number of iterations of Algorithm 2 with dependencies on all the algorithm hyperparameters can be found in Eq. (74) in the proofs.*

*(c) The general research on complexity analysis for optimization algorithms under local Lipschitz assumptions is relatively new. For example, Lu & Mei (2023) proposes accelerated gradient methods for convex optimization problems with locally Lipschitz continuous gradients, and Zhang & Hong (2024) proposes accelerated gradient methods for nonconvex optimization problems with locally Lipschitz continuous gradients.*

## 4.3 Communication overheads

In each iteration of Algorithm 2, a single communication round happens between the clients and the central server. During this round, the central server transmits the global weight $w^{t+1}$ to all clients, and subsequently each local client performs multiple local updates to solve a local subproblem and then sends the updated local weights $\tilde{u}_i^{t+1}$ and a local stationarity measure $\tilde{\varepsilon}_{i,t+1}$ back to the central server. The communication complexity of each call of Algorithm 2 is $\mathcal{O}(|\log \tau|)$.

# 5 Numerical experiments

Here, we conduct numerical experiments to evaluate the performance of our proposed FL algorithm (Algorithm 1). Specifically, we benchmark our algorithm against a centralized proximal AL method (cProx-AL, described in Algorithm 3) on a convex Neyman-Pearson classification problem (Section 5.1) and a fair-aware learning problem (Section 5.2) with real-world datasets, and further on linear-equality-constrained quadratic programming problems with simulated data (Appendix E.2). All experiments are carried out on a Windows system with an AMD EPYC 7763 64-core processor, and all algorithms are implemented in Python. The code to implement the proposed algorithm on these numerical examples is available at https://github.com/PL97/Constr_FL.

Table 1: Numerical results for solving Eq. (30) using our algorithm vs. using cProx-AL. Inside the parentheses are the respective standard deviations over 10 random trials. For feasibility, we include the mean and maximum losses for class 1 among all local clients.

| dataset | $n$ | objective value (loss for class 0) | | | feasibility (loss for class 1 ($\leq 0.2$)) | | | |
| | | Algorithm 1 | cProx-AL | relative difference | Algorithm 1 | | cProx-AL | |
| | | | | | mean | max | mean | max |
|---|---|---|---|---|---|---|---|---|
| breast-cancer-wisc | 1 | 0.27 (1.52e-04) | 0.27 (3.02e-05) | 7.09e-04 (2.02e-04) | 0.20 (1.80e-07) | 0.20 (1.80e-07) | 0.20 (1.84e-08) | 0.20 (1.84e-08) |
| | 5 | 0.34 (4.50e-02) | 0.33 (4.55e-02) | 1.15e-02 (5.17e-03) | 0.19 (7.33e-06) | 0.20 (1.08e-06) | 0.19 (1.13e-04) | 0.20 (1.72e-05) |
| | 10 | 0.37 (1.08e-01) | 0.37 (1.08e-01) | 3.92e-04 (2.76e-04) | 0.17 (1.15e-05) | 0.20 (6.05e-09) | 0.17 (1.14e-05) | 0.20 (2.95e-08) |
| | 20 | 0.46 (2.12e-01) | 0.45 (2.12e-01) | 3.43e-02 (2.91e-02) | 0.16 (3.52e-05) | 0.20 (3.76e-06) | 0.16 (7.03e-06) | 0.20 (7.70e-08) |
| adult-a | 1 | 0.73 (2.19e-04) | 0.73 (1.25e-04) | 2.24e-04 (3.46e-04) | 0.20 (6.30e-07) | 0.20 (6.30e-07) | 0.20 (1.73e-06) | 0.20 (1.73e-06) |
| | 5 | 0.74 (1.03e-02) | 0.74 (1.03e-02) | 4.25e-03 (7.44e-04) | 0.20 (2.14e-04) | 0.20 (2.80e-04) | 0.20 (1.21e-05) | 0.20 (2.28e-06) |
| | 10 | 0.77 (1.98e-02) | 0.77 (1.98e-02) | 2.69e-03 (3.24e-03) | 0.19 (6.41e-05) | 0.20 (9.76e-05) | 0.19 (2.00e-05) | 0.20 (1.23e-05) |
| | 20 | 0.78 (2.86e-02) | 0.79 (2.81e-02) | 1.13e-02 (4.11e-03) | 0.18 (6.40e-04) | 0.20 (6.59e-05) | 0.18 (1.96e-05) | 0.20 (3.19e-06) |
| monks-1 | 1 | 1.58 (7.61e-05) | 1.58 (7.50e-05) | 1.39e-05 (1.09e-05) | 0.20 (1.09e-07) | 0.20 (1.09e-07) | 0.20 (3.01e-07) | 0.20 (3.01e-07) |
| | 5 | 1.65 (8.39e-02) | 1.65 (8.41e-02) | 2.08e-04 (1.84e-04) | 0.19 (6.39e-05) | 0.20 (5.39e-05) | 0.19 (5.04e-06) | 0.20 (5.60e-07) |
| | 10 | 1.71 (1.18e-01) | 1.71 (1.18e-01) | 4.59e-04 (3.32e-04) | 0.18 (3.98e-05) | 0.20 (4.46e-05) | 0.18 (6.44e-06) | 0.20 (1.60e-06) |
| | 20 | 1.81 (1.49e-01) | 1.79 (1.60e-01) | 1.78e-02 (1.38e-02) | 0.17 (1.68e-04) | 0.20 (2.24e-04) | 0.17 (4.60e-06) | 0.20 (1.62e-06) |

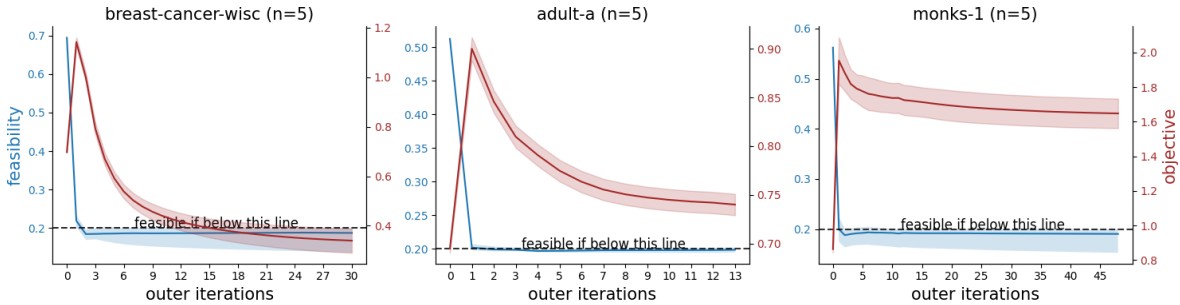

Figure 1: Convergence behavior of local objective and local feasibility in one random trial over the outer iterations of Algorithm 1 on three real-world datasets. The solid blue and brown lines indicate the mean local objective and the mean local feasibility over all clients, respectively. The blue and the brown areas indicate the cross-client variations of local objectives and local feasibility, respectively. The dashed black line indicates the feasibility threshold.

## 5.1 Neyman-Pearson classification

In this subsection, we consider the Neyman-Pearson classification problem:

$$\min_w \frac{1}{n} \sum_{i=1}^{n} \frac{1}{m_{i0}} \sum_{j=1}^{m_{i0}} \phi(w; (x_j^{(i0)}, 0)) \quad \text{s.\,t.} \quad \frac{1}{m_{i1}} \sum_{j=1}^{m_{i1}} \phi(w; (x_j^{(i1)}, 1)) \leq r_i, \quad 1 \leq i \leq n, \tag{30}$$

where $\{x_j^{(i0)}\}_{1 \leq j \leq m_{i0}}$ and $\{x_j^{(i1)}\}_{1 \leq j \leq m_{i1}}$ are the sets of samples at client $i$ associated with labels 0 and 1, respectively, and $\phi$ is the binary logistic loss (Hastie et al., 2009)

$$\phi(w; (x, y)) = -y w^T x + \log(1 + e^{w^T x}), \quad y \in \{0, 1\}. \tag{31}$$

Then, both the objective and the constraints in Eq. (30) are convex. We consider three real-world datasets, namely 'breast-cancer-wisc', 'adult-a', and 'monks-1', from the UCI repository[2] and described in Appendix E.1. For each dataset, we perform the Neyman-Pearson classification that minimizes the loss of classification for class 0 (majority) while ensuring that the loss for class 1 (minority) is less than a threshold $r_i = 0.2$. To simulate the FL setting, we divide each dataset into $n$ folds, mimicking local clients, each holding the same amount of data with equal ratios of the two classes.

We apply Algorithm 1 and cProx-AL (Algorithm 3) to find a $(10^{-3}, 10^{-3})$-optimal solution of Eq. (30). We run 10 trials of Algorithm 1 and cProx-AL. For each run, both algorithms have the same initial point $w^0$,

---

[2]see https://archive.ics.uci.edu/datasets

randomly chosen from the unit Euclidean sphere. We set the other parameters for Algorithm 1 and cProx-AL as $\mu_i^0 = (0, \ldots, 0)^T \ \forall 0 \leq i \leq n$, $\bar{s} = 0.001$ and $\beta = 300$. We also set $\rho_i = 0.01 \ \forall 1 \leq i \leq n$ for Algorithm 2.

Comparing the objective value and feasibility of solutions achieved by Algorithm 1 and cProx-AL in Table 1, we see that both algorithms can yield solutions of similar quality. Given the small standard deviations, we observe that the convergence behavior of Algorithm 1 remains stable across 10 trial runs. These observations demonstrate the ability of Algorithm 1 to reliably solve Eq. (30) in the FL setting without compromising solution quality. From Fig. 1, we observe that Algorithm 1 consistently achieves feasibility for all local constraints while also minimizing all the local objectives.

## 5.2 Classification with fairness constraints

In this subsection, we consider fairness-aware learning with global and local fairness constraints:

$$\min_w \frac{1}{n} \sum_{i=1}^n \frac{1}{m_i} \sum_{j=1}^{m_i} \phi(w; z_j^{(i)}) \quad \text{s.t.} \quad -r_i \leq \frac{1}{\tilde{m}_i} \sum_{j=1}^{\tilde{m}_i} \phi(w; \tilde{z}_j^{(i)}) - \frac{1}{\hat{m}_i} \sum_{j=1}^{\hat{m}_i} \phi(w; \hat{z}_j^{(i)}) \leq r_i, \quad 0 \leq i \leq n. \quad (32)$$

Here, $\{z_j^{(i)} = (x_j^{(i)}, y_j^{(i)}) \in \mathbb{R}^d \times \{0, 1\} : i = 0, \ldots, n, j = 1, \ldots, m_i\}$ is the training set, where $i$ indexes the central server/local clients. For each $i = 0, \ldots, n$, the dataset $\{z_j^{(i)}\}_{1 \leq j \leq m_i}$ is further divided into two subgroups $\{\tilde{z}_j^{(i)}\}_{1 \leq j \leq \tilde{m}_i}$ and $\{\hat{z}_j^{(i)}\}_{1 \leq j \leq \hat{m}_i}$ based on certain subgroup attributes. The constraints with $i = 1, \ldots, n$ refer to local constraints at client $i$, while the constraints with $i = 0$ refer to global constraints at the central server.

We choose $\phi$ as the binary logistic loss defined in Eq. (31), leading to nonconvex constraints in Eq. (32). For the real-world dataset, we consider 'adult-b'[3]: each sample in this dataset has 39 features and one binary label. To simulate the FL setting, we divide the $22,654$ training samples from the 'adult-b' dataset into $n$ folds and distribute them to $n$ local clients. The central server holds the $5,659$ test samples from the 'adult-b' dataset. Note that although we have taken both the "training" and "test" samples from the 'adult-b' dataset here, these samples are used to simulate our local samples and central samples, respectively. The focus here is to test optimization performance, not generalization—we do not have a test step, unlike in typical supervised learning.

We apply Algorithm 1 and cProx-AL (Algorithm 3) to find a $(10^{-3}, 10^{-3})$-optimal solution of Eq. (32). We run 10 trials of Algorithm 1 and cProx-AL. For each run, both algorithms have the same initial point $w^0$, randomly chosen from the unit Euclidean sphere. We set the other parameters for Algorithm 1 and cProx-AL as $\mu_i^0 = (0, \ldots, 0)^T \ \forall 0 \leq i \leq n$, $\bar{s} = 0.001$ and $\beta = 10$. We also set $\rho_i = 10^8 \ \forall 1 \leq i \leq n$ for Algorithm 2.

Comparing the objective value and feasibility of solutions achieved by Algorithm 1 and cProx-AL in Table 2 reveals that Algorithm 1 and cProx-AL can produce solutions of similar quality. Given the small standard deviations, we observe that the convergence behavior of Algorithm 1 remains stable across 10 trial runs. These observations demonstrate the ability of Algorithm 1 to reliably solve Eq. (32) in the FL setting without compromising solution quality. It also suggests the potential of our algorithm in solving FL problems with nonconvex constraints. From Fig. 2, we see that our proposed method consistently achieves feasibility for all local and global constraints while also minimizing all the local objectives.

## 6 Concluding remarks

In this paper, we propose an FL algorithm for solving general constrained ML problems based on the proximal AL method. We analyze the worst-case iteration complexity of the proposed algorithm, assuming convex objective and convex constraints with locally Lipschitz continuous gradients. Finally, we perform numerical experiments to assess the performance of the proposed algorithm for constrained classification problems, using real-world datasets. The numerical results clearly demonstrate the practical efficacy of our proposed algorithm. Since our work is the first of its kind, there are numerous possible future directions. For example,

---

[3]This dataset can be found in `https://github.com/heyaudace/ml-bias-fairness/tree/master/data/adult`.

Table 2: Numerical results for Eq. (32) using our algorithm vs. using cProx-AL. Inside the parentheses are the respective standard deviations over 10 random trials. For feasibility, we include the mean and maximum loss disparities (absolute difference between losses for two subgroups) among all clients and the central server.

| | objective value | | | feasibility (loss disparity ($\leq 0.1$)) | | | |
| $n$ | Algorithm 1 | cProx-AL | relative difference | Algorithm 1 | | cProx-AL | |
| | | | | mean | max | mean | max |
|---|---|---|---|---|---|---|---|
| 1 | 0.37 (9.83e-05) | 0.37 (4.14e-05) | 1.97e-03 (2.53e-04) | 0.10 (1.14e-04) | 0.10 (1.36e-04) | 0.10 (3.69e-06) | 0.10 (5.38e-06) |
| 5 | 0.37 (3.99e-03) | 0.37 (4.05e-03) | 1.86e-03 (4.69e-04) | 0.09 (5.34e-05) | 0.10 (7.51e-05) | 0.09 (3.68e-05) | 0.10 (4.36e-06) |
| 10 | 0.37 (6.39e-03) | 0.37 (6.52e-03) | 2.39e-03 (8.40e-04) | 0.08 (1.68e-04) | 0.10 (2.15e-05) | 0.08 (1.52e-04) | 0.10 (6.56e-06) |
| 20 | 0.38 (9.46e-03) | 0.37 (9.86e-03) | 4.61e-03 (2.43e-03) | 0.08 (9.75e-05) | 0.10 (1.01e-04) | 0.08 (4.90e-05) | 0.10 (6.06e-06) |

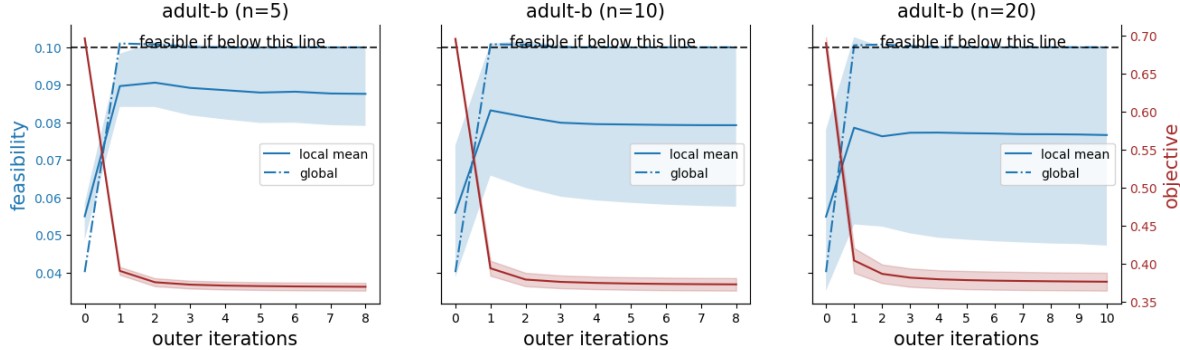

Figure 2: Convergence of local objective, local feasibility, and the feasibility for global constraints in one random trial over the outer iterations of Algorithm 1. The solid blue and brown lines indicate the mean local objective and the mean local feasibility over all clients, respectively. The blue and brown areas indicate the cross-client variations of local objectives and local feasibility, respectively. The dashdot blue line indicates the feasibility for global constraints. The dashed black line indicates the feasibility threshold.

one could try to extend our FL algorithms to allow partial client participation and stochastic solvers at local clients. In addition, developing FL algorithms for general constrained ML with convergence guarantees in nonconvex settings remains largely open. Lastly, constrained FL with a fixed iteration and communication budget, especially stringent ones, is a very useful but challenging future research topic.

**Acknowledgments**

C. He is partially supported by the NIH fund R01CA287413 and the UMN Research Computing Seed Grant. L. Peng is partially supported by the CISCO Research fund 1085646 PO USA000EP390223. J. Sun is partially supported by the NIH fund R01CA287413 and the CISCO Research fund 1085646 PO USA000EP390223. The authors acknowledge the Minnesota Supercomputing Institute (MSI) at the University of Minnesota for providing resources that contributed to the research results reported in this article. The content is solely the responsibility of the authors and does not necessarily represent the official views of the National Institutes of Health.

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

## Appendix

In [Appendices A](#) to [C](#), we provide proofs of the main results in [Sections 3](#) and [4](#). [Appendix D](#) presents a proximal AL method for centralized constrained optimization. In [Appendix E](#), we include some extra numerical results.

## A  Proofs of [Theorem 3.1](#), [Lemma 3.1](#), and [Theorem 3.2](#)(a)

First, we set up the technical tools necessary for the proof, following [Lu & Zhou (2023)](#). With the abbreviations in [Eq. (6)](#), we define the Lagrangian function associated with [Eqs. (5)](#) and [(7)](#) as

$$
l(w, \mu) = \begin{cases} f(w) + h(w) + \langle \mu, c(w) \rangle & \text{if } w \in \text{dom}(h) \text{ and } \mu \geq 0, \\ -\infty & \text{if } w \in \text{dom}(h) \text{ and } \mu \ngeq 0, \\ \infty & \text{if } w \notin \text{dom}(h), \end{cases}
$$

Then, one can verify that

$$
\partial l(w, \mu) = \begin{cases} \begin{pmatrix} \nabla f(w) + \partial h(w) + \nabla c(w)\mu \\ c(w) - \mathcal{N}_{\mathbb{R}^m_+}(\mu) \end{pmatrix} & \text{if } w \in \text{dom}(h) \text{ and } \mu \geq 0, \\ \emptyset & \text{otherwise.} \end{cases} \tag{33}
$$

We also define a set-valued operator $\mathcal{T}$ associated with [Eqs. (5)](#) and [(7)](#):

$$
\mathcal{T} : (w, \mu) \to \{(u, \nu) \in \mathbb{R}^d \times \mathbb{R}^m : (u, -\nu) \in \partial l(w, \mu)\}, \quad \forall (w, \mu) \in \mathbb{R}^d \times \mathbb{R}^m, \tag{34}
$$

which is maximally monotone (see, e.g., Section 2 of [Rockafellar (1976)](#)). Finding a KKT solution of [Eq. (5)](#) can be viewed as solving the monotone inclusion problem ([Rockafellar, 1976](#)):

$$
\text{Find} \quad (w, \mu) \in \mathbb{R}^d \times \mathbb{R}^m \quad \text{such that} \quad (0, 0) \in \mathcal{T}(w, \mu). \tag{35}
$$

Furthermore, applying the proximal AL method to solve [Eq. (5)](#) is equivalent to applying the proximal point algorithm (PPA) to solve this monotone inclusion problem ([Rockafellar, 1976](#); [Bauschke & Combettes, 2017](#)), that is,

$$
w^{k+1} = \arg\min_w \ell_k(w), \quad \mu^{k+1} = [\mu^k + \beta c(w^{k+1})]_+, \iff (w^{k+1}, \mu^{k+1}) = \mathcal{J}(w^k, \mu^k), \quad \forall k \geq 0, \tag{36}
$$

where $(w^0, \mu^0) \in \text{dom}(h) \times \mathbb{R}^m_+$ and $\mathcal{J}$ is the resolvent of $\mathcal{T}$ defined as

$$
\mathcal{J} := (\mathcal{I} + \beta \mathcal{T})^{-1} \tag{37}
$$

with $\mathcal{I}$ being the identity operator. When the $\arg\min_w \ell_k(w)$ subproblem is only solved up to approximate stationarity, that is, $\text{dist}_\infty(0, \partial \ell_k(w^{k+1})) \leq \tau_k$ as in our [Algorithm 1](#), the error $\tau_k$ will propagate to the next iterate that we obtain. This is quantitatively captured by the following result.

**Lemma A.1 (adaptation of Lemma 5 of [Lu & Zhou (2023)](#)).** *Suppose that Assumptions [0](#) to [2](#) hold. Let $\{(w^k, \mu^k)\}_{k \geq 0}$ be generated by [Algorithm 1](#). Then for any $k \geq 0$, we have*

$$
\|(w^{k+1}, \mu^{k+1}) - \mathcal{J}(w^k, \mu^k)\| \leq \beta\sqrt{n}\tau_k,
$$

*where $\mathcal{J}$ is the resolvent of $\mathcal{T}$ defined in [Eq. (37)](#).*

*Proof.* Notice from [Eq. (10)](#) that $\text{dist}(0, \partial \ell_k(w^{k+1})) \leq \sqrt{n}\text{dist}_\infty(0, \partial \ell_k(w^{k+1})) \leq \sqrt{n}\tau_k$. By this and Lemma 5 of [Lu & Zhou (2023)](#), the conclusion of this lemma holds. □

### A.1 Proof of Theorem 3.1

*Proof of Theorem 3.1.* Notice from Eqs. (6) and (9) that

$$\ell_k(w) = f(w) + h(w) + \frac{1}{2\beta} \left( \|[\mu^k + \beta c(w)]_+\|^2 - \|\mu^k\|^2 \right) + \frac{1}{2\beta}\|w - w^k\|^2.$$

By this, Eq. (33), and the fact that $\mu^{k+1} = [\mu^k + \beta c(w^{k+1})]_+$, one has

$$\partial \ell_k(w^{k+1}) - \frac{1}{\beta}(w^{k+1} - w^k) = \nabla f(w^{k+1}) + \partial h(w^{k+1}) + \nabla c(w^{k+1})[\mu^k + \beta c(w^{k+1})]_+$$

$$= \nabla f(w^{k+1}) + \partial h(w^{k+1}) + \nabla c(w^{k+1})\mu^{k+1} = \partial_w l(w^{k+1}, \mu^{k+1}). \quad (38)$$

Notice that

$$\mu^{k+1} = [\mu^k + \beta c(w^{k+1})]_+ = \underset{\mu \in \mathbb{R}_+^m}{\arg\min} \frac{1}{2}\|\mu - (\mu^k + \beta c(w^{k+1}))\|^2.$$

By the optimality condition of this projection, we have

$$0 \in \mu^{k+1} - (\mu^k + \beta c(w^{k+1})) + \mathcal{N}_{\mathbb{R}_+^m}(\mu^{k+1}),$$

which together with Eq. (33) implies that

$$\frac{1}{\beta}(\mu^{k+1} - \mu^k) \in \partial_\mu l(w^{k+1}, \mu^{k+1}). \quad (39)$$

In view of this, Eqs. (10), (11) and (38), we can see that

$$\text{dist}_\infty(0, \partial_w l(w^{k+1}, \mu^{k+1})) \overset{Eq.\ (38)}{\leq} \text{dist}_\infty(0, \partial \ell_k(w^{k+1})) + \frac{1}{\beta}\|w^{k+1} - w^k\|_\infty$$

$$\overset{Eq.\ (10)}{\leq} \tau_k + \frac{1}{\beta}\|w^{k+1} - w^k\|_\infty \overset{Eq.\ (11)}{\leq} \epsilon_1,$$

$$\text{dist}_\infty(0, \partial_\mu l(w^{k+1}, \mu^{k+1})) \overset{Eq.\ (39)}{\leq} \frac{1}{\beta}\|\mu^{k+1} - \mu^k\|_\infty \overset{Eq.\ (11)}{\leq} \epsilon_2.$$

These along with Eq. (33) and Definition 1 imply that $(w^{k+1}, \mu^{k+1})$ is an $(\epsilon_1, \epsilon_2)$-KKT solution of Eq. (5), which proves this theorem as desired. $\qquad \square$

### A.2 Proof of Lemma 3.1

Define

$$w_*^k := \underset{w}{\arg\min}\, \ell_k(w), \qquad \mu_*^k := [\mu^k + \beta c(w_*^k)]_+, \quad \forall k \geq 0, \quad (40)$$

which, by Eq. (36), is equivalent to

$$(w_*^k, \mu_*^k) = \mathcal{J}(w^k, \mu^k). \quad (41)$$

Recall that $(w^*, \mu^*)$ is assumed to be any pair of optimal solutions to Eq. (5) and Eq. (7). Toward the proof, we first present an intermediate result, which mostly follows the fact that $\mathcal{J}$ is firmly nonexpansive.

**Lemma A.2.** *Suppose that Assumptions 0 to 2 hold. Let $\{(w^k, \mu^k)\}_{k \geq 0}$ be generated by Algorithm 1. Let $(w_*^k, \mu_*^k)$ be defined in Eq. (40) for all $k \geq 0$. Then the following relations hold.*

$$\|(w^k, \mu^k) - (w_*^k, \mu_*^k)\|^2 + \|(w_*^k, \mu_*^k) - (w^*, \mu^*)\|^2 \leq \|(w^k, \mu^k) - (w^*, \mu^*)\|^2, \quad \forall k \geq 0, \quad (42)$$

$$\|(w^k, \mu^k) - (w^*, \mu^*)\| \leq \|(w^0, \mu^0) - (w^*, \mu^*)\| + \beta\sqrt{n}\sum_{j=0}^{k-1}\tau_j, \quad \forall k \geq 0. \quad (43)$$

*Proof.* Since $(w^*, \mu^*)$ is a solution to the monotone inclusion problem Eq. (35), we have

$$(0,0) \in \mathcal{T}(w^*, \mu^*), \quad \text{and} \quad (w^*, \mu^*) = \mathcal{J}(w^*, \mu^*). \tag{44}$$

Moreover, since $\mathcal{T}$ is maximally monotone, its resolvent $\mathcal{J}$ is firmly nonexpansive (see, e.g., Corollary 23.9 of Bauschke & Combettes (2017)), that is, $\| \mathcal{J}(w, \mu) - \mathcal{J}(w', \mu') \|^2 + \| (\mathcal{I} - \mathcal{J})(w, \mu) - (\mathcal{I} - \mathcal{J})(w', \mu') \|^2 \leq \| (w, \mu) - (w', \mu') \|^2$ for any feasible pairs $(w, \mu)$ and $(w', \mu')$. Using Eqs. (41) and (44), we obtain that

$$\| (w^k, \mu^k) - (w_*^k, \mu_*^k) \|^2 + \| (w_*^k, \mu_*^k) - (w^*, \mu^*) \|^2$$
$$\overset{Eqs. (41) and (44)}{=} \| (\mathcal{I} - \mathcal{J})(w^k, \mu^k) - (\mathcal{I} - \mathcal{J})(w^*, \mu^*) \|^2 + \| \mathcal{J}(w^k, \mu^k) - \mathcal{J}(w^*, \mu^*) \|^2$$
$$\leq \| (w^k, \mu^k) - (w^*, \mu^*) \|^2. \quad \text{(firm nonexpansiveness of } \mathcal{J})$$

Hence, Eq. (42) holds as desired.

Now we prove Eq. (43). It suffices to consider the case where $k \geq 1$. We have

$$\| (w^k, \mu^k) - (w^*, \mu^*) \| \leq \| (w^k, \mu^k) - \mathcal{J}(w^{k-1}, \mu^{k-1}) \| + \| \mathcal{J}(w^{k-1}, \mu^{k-1}) - \mathcal{J}(w^*, \mu^*) \|$$
$$\leq \beta \sqrt{n} \tau_{k-1} + \| (w^{k-1}, \mu^{k-1}) - (w^*, \mu^*) \|, \tag{45}$$

where we have invoked Lemma A.1 and the nonexpansiveness of $\mathcal{J}$ to obtain the final upper bound. Repeatedly applying Eq. (45) for iterates $(w^1, \mu^1)$ through $(w^{k-1}, \mu^{k-1})$, we have

$$\beta \sqrt{n} \tau_{k-1} + \| (w^{k-1}, \mu^{k-1}) - (w^*, \mu^*) \| \leq \beta \sqrt{n} \sum_{j=0}^{k-1} \tau_j + \| (w^0, \mu^0) - (w^*, \mu^*) \|, \tag{46}$$

completing the proof. $\qquad \square$

*Proof of Lemma 3.1.* Notice from Algorithm 1 that $\tau_k = \bar{s}/(k+1)^2$ for all $k \geq 0$. Therefore, one has $\sum_{j=0}^{\infty} \tau_j \leq 2\bar{s}$. In view of this, Eq. (15), and Lemma A.2, we observe that

$$\max\{ \| w^k - w^* \|, \| \mu^k - \mu^* \|, \| w^k - w_*^k \|, \| w_*^k - w^* \| \} \leq r_0 + 2\sqrt{n} \bar{s} \beta, \quad \forall k \geq 0. \tag{47}$$

where $r_0$ is defined in Eq. (15), and $\beta$ and $\bar{s}$ are inputs of Algorithm 1. Eq. (47) implies that $w^k \in \mathcal{Q}_1$ for all $k \geq 0$, completing the proof. $\qquad \square$

## A.3  Proof of Theorem 3.2(a)

To prove Theorem 3.2(a), we first present a general technical lemma on the convergence rate of an inexact PPA applied to monotone inclusion problems.

**Lemma A.3** (**restatement of Lemma 3 of Lu & Zhou (2023)**). *Let $\widetilde{\mathcal{T}} : \mathbb{R}^p \rightrightarrows \mathbb{R}^q$ be a maximally monotone operator and $z^* \in \mathbb{R}^p$ such that $0 \in \widetilde{\mathcal{T}}(z^*)$. Let $\{z^k\}$ be a sequence generated by an inexact PPA, starting with $z^0$ and obtaining $z^{k+1}$ by approximately evaluating $\widetilde{\mathcal{J}}(z^k)$ such that*

$$\| z^{k+1} - \widetilde{\mathcal{J}}(z^k) \| \leq e_k$$

*for some $\beta > 0$ and $e_k \geq 0$, where $\widetilde{\mathcal{J}} := (\mathcal{I} + \beta\widetilde{\mathcal{T}})^{-1}$ and $\mathcal{I}$ is the identity operator. Then, for any $K \geq 1$, we have*

$$\min_{K \leq k \leq 2K} \| z^{k+1} - z^k \| \leq \frac{\sqrt{2} \left( \| z^0 - z^* \| + 2 \sum_{k=0}^{2K} e_k \right)}{\sqrt{K+1}}.$$

*Proof of Theorem 3.2(a).* Observe that Algorithm 1 terminates when two consecutive iterates $(w^{k+1}, \mu^{k+1})$ and $(w^k, \mu^k)$ are close. We use this observation and Lemmas A.1 and A.3 to derive the maximum number of outer iterations of Algorithm 1.

Recall that $\sum_{j=0}^{\infty} \tau_j \leq 2\bar{s}$. It follows from Lemmas A.1 and A.3 that

$$\min_{K \leq k \leq 2K} \frac{1}{\beta} \|(w^{k+1}, \mu^{k+1}) - (w^k, \mu^k)\| \leq \frac{\sqrt{2} \left( \|(w^0, \mu^0) - (w^*, \mu^*)\| + 2\sqrt{n}\beta \sum_{j=0}^{\infty} \tau_j \right)}{\beta\sqrt{K+1}}$$

$$\leq \frac{\sqrt{2} \left( \|(w^0, \mu^0) - (w^*, \mu^*)\| + 4\sqrt{n}\bar{s}\beta \right)}{\beta\sqrt{K+1}} = \frac{\sqrt{2} \left( r_0 + 4\sqrt{n}\bar{s}\beta \right)}{\beta\sqrt{K+1}},$$

which then implies that

$$\min_{K \leq k \leq 2K} \left\{ \tau_k + \frac{1}{\beta} \|w^{k+1} - w^k\|_\infty \right\} \leq \frac{\bar{s}}{(K+1)^2} + \frac{\sqrt{2} \left( r_0 + 4\sqrt{n}\bar{s}\beta \right)}{\beta\sqrt{K+1}} \leq \left[ \bar{s} + \frac{\sqrt{2} \left( r_0 + 4\sqrt{n}\bar{s}\beta \right)}{\beta} \right] \frac{1}{\sqrt{K+1}},$$

$$\min_{K \leq k \leq 2K} \frac{1}{\beta} \|\mu^{k+1} - \mu^k\|_\infty \leq \frac{\sqrt{2} \left( r_0 + 4\sqrt{n}\bar{s}\beta \right)}{\beta\sqrt{K+1}}.$$

We see from these and the termination criterion in Eq. (11) that the number of outer iterations of Algorithm 1 is at most

$$K_{\epsilon_1, \epsilon_2} := \left[ \bar{s} + \frac{\sqrt{2}(r_0 + 4\sqrt{n}\bar{s}\beta)}{\beta} \right]^2 \max\{\epsilon_1^{-2}, \epsilon_2^{-2}\} = \mathcal{O}(\max\{\epsilon_1^{-2}, \epsilon_2^{-2}\}). \tag{48}$$

Hence, Theorem 3.2(a) holds as desired. $\qquad\square$

## B   Proofs of the main results in Section 4

Throughout this section, we let $(\tilde{w}^*, u^*)$ be the optimal solution of Eq. (24), and $\lambda^*$ be the associated Lagrangian multiplier. Recall from the definition of $\tilde{u}_i^0$ in Algorithm 2 and Eq. (21) that

$$\tilde{u}_i^t = u_i^t + \lambda_i^t/\rho_i, \quad \forall 1 \leq i \leq n, t \geq 0. \tag{49}$$

### B.1   Proof of Lemma 4.1

For notational convenience, write $f_0(w) \equiv 0$. Then, by Eqs. (13) and (14), one can verify that

$$\nabla P_{i,k}(w) = \nabla f_i(w) + \nabla c_i(w)[\mu_i^k + \beta c_i(w)]_+ + \frac{1}{(n+1)\beta}(w - w^k), \quad \forall 0 \leq i \leq n. \tag{50}$$

*Proof of Lemma 4.1.* Fix an arbitrary $w \in \mathbb{R}^d$ and a bounded open set $\mathcal{U}_w$ containing $w$. We suppose that $\nabla f_i$ is $L_{w,1}$-Lipschitz continuous on $\mathcal{U}_w$, and $\nabla c_i$ is $L_{w,2}$-Lipschitz continuous on $\mathcal{U}_w$. Also, let $U_{w,1} = \sup_{w \in \mathcal{U}_w} \|c_i(w)\|$ and $U_{w,2} = \sup_{w \in \mathcal{U}_w} \|\nabla c_i(w)\|$. By Eqs. (13), (14) and (50)) one has for each $0 \leq i \leq n$ and $u, v \in \mathcal{U}_w$ that

$$\|\nabla P_{i,k}(u) - \nabla P_{i,k}(v)\| \overset{Eq.\ (50)}{\leq} \|\nabla f_i(u) - \nabla f_i(v)\| + \|\nabla c_i(u) - \nabla c_i(v)\|\|[\mu_i^k + \beta c_i(u)]_+\|$$

$$+ \|[\mu_i^k + \beta c_i(u)]_+ - [\mu_i^k + \beta c_i(v)]_+\|\|\nabla c_i(v)\| + \frac{1}{(n+1)\beta}\|u - v\|$$

$$\leq L_{w,1}\|u - v\| + (\|\mu_i^k\| + \beta U_{w,1})L_{w,2}\|u - v\|$$

$$+ \beta\|c_i(u) - c_i(v)\|\|\nabla c_i(v)\| + \frac{1}{(n+1)\beta}\|u - v\|$$

$$\leq \left[ L_{w,1} + (\|\mu_i^k\| + \beta U_{w,1})L_{w,2} + \beta U_{w,2}^2 + \frac{1}{(n+1)\beta} \right] \|u - v\|.$$

Therefore, $\nabla P_{i,k}(u)$ is locally Lipschitz continuous on $\mathbb{R}^d$, and the conclusion holds as desired. $\qquad\square$

## B.2 Proof of Theorem 4.1

*Proof of Theorem 4.1.* In view of the termination criterion Eq. (23), it suffices to show that

$$\text{dist}_\infty(0, \partial\ell(w^{T+1})) \leq \varepsilon_{T+1} + \sum_{i=1}^n \tilde{\varepsilon}_{i,T+1}.$$

By the definition of $\ell$ in Eq. (16), one has that

$$\partial\ell(w^{T+1}) = \sum_{i=0}^n \nabla P_i(w^{T+1}) + \partial h(w^{T+1}). \tag{51}$$

In addition, notice from Eqs. (18), (19) and (49) that

$$\partial\varphi_{0,T}(w^{T+1}) = \nabla P_0(w^{T+1}) + \sum_{i=1}^n \rho_i(w^{T+1} - \tilde{u}_i^T) + \partial h(w^{t+1})$$

$$= \nabla P_0(w^{T+1}) + \sum_{i=1}^n [\rho_i(w^{T+1} - u_i^T) - \lambda_i^T] + \partial h(w^{T+1}),$$

$$\nabla\varphi_{i,T}(w^{T+1}) = \nabla P_i(w^{T+1}) + \lambda_i^T, \quad \forall 1 \leq i \leq n.$$

Combining these with Eq. (51), we obtain that

$$\partial\ell(w^{T+1}) = \partial\varphi_{0,T}(w^{T+1}) + \sum_{i=1}^n [\nabla\varphi_{i,T}(w^{T+1}) - \rho_i(w^{T+1} - u_i^T)],$$

which together with $\text{dist}_\infty(0, \partial\varphi_{0,T}(w^{T+1})) \leq \varepsilon_{T+1}$ (see Algorithm 2 and Eq. (22)) implies that

$$\text{dist}_\infty(0, \partial\ell(w^{T+1})) \leq \text{dist}_\infty(0, \partial\varphi_{0,T}(w^{T+1})) + \sum_{i=1}^n \|\nabla\varphi_{i,T}(w^{T+1}) - \rho_i(w^{T+1} - u_i^T)\|_\infty$$

$$\leq \varepsilon_{T+1} + \sum_{i=1}^n \tilde{\varepsilon}_{i,T+1},$$

as desired. □

## B.3 Proof of Lemma 4.2

To prove Lemma 4.2, we use convergence analysis techniques for ADMM to show that the distances between iterates $\{u_i^k\}_{1 \leq i \leq n}$ and $w^k$ and the optimal solution $\tilde{w}^*$ are controlled by the distance between the initial iterate $\tilde{w}^0$ and $\tilde{w}^*$. To the best of our knowledge, such boundedness results without assuming global Lipschitz continuity are entirely new in the literature on ADMM.

*Proof of Lemma 4.2.* From the optimality conditions and stopping criteria for Eq. (18) and Eq. (19), there exist $e_i^{t+1}$'s for $0 \leq i \leq n$ with $\|e_i^{t+1}\|_\infty \leq \varepsilon_{t+1}$ and $h^{t+1} \in \partial h(w^{t+1})$ so that:

$$e_0^{t+1} = \nabla P_0(w^{t+1}) + h^{t+1} + \sum_{i=1}^n \rho_i(w^{t+1} - \tilde{u}_i^t) \overset{Eq.~(49)}{=} \nabla P_0(w^{t+1}) + h^{t+1} + \sum_{i=1}^n [\rho_i(w^{t+1} - u_i^t) - \lambda_i^t]$$

$$\overset{Eq.~(20)}{=} \nabla P_0(w^{t+1}) + h^{t+1} + \sum_{i=1}^n [\rho_i(u_i^{t+1} - u_i^t) - \lambda_i^{t+1}] \tag{52}$$

and

$$e_i^{t+1} = \nabla\varphi_{i,t}(u_i^{t+1}) \overset{Eq.~(19)}{=} \nabla P_i(u_i^{t+1}) + \lambda_i^t + \rho_i(u_i^{t+1} - w^{t+1}) \overset{Eq.~(20)}{=} \nabla P_i(u_i^{t+1}) + \lambda_i^{t+1}, \quad \forall 1 \leq i \leq n. \tag{53}$$

Moreover, since $\tilde{w}^*$ and $u^*$ are the optimal solution of Eq. (24) with the associated Lagrangian multiplier $\lambda^* \in \mathbb{R}^m$, we have by the optimality condition that there exists $h^* \in \partial h(\tilde{w}^*)$ such that

$$\nabla P_i(u_i^*) + \lambda_i^* = 0, \quad \nabla P_0(\tilde{w}^*) + h^* - \sum_{i=1}^{n} \lambda_i^* = 0, \quad u_i^* = \tilde{w}^*, \quad \forall 1 \leq i \leq n. \tag{54}$$

Recall that $P_i$, $0 \leq i \leq n$, are strongly convex with the modulus $\sigma > 0$, we have

$$\begin{aligned}
\sigma \|u_i^{t+1} - \tilde{w}^*\|^2 &\leq \langle u_i^{t+1} - \tilde{w}^*, \nabla P_i(u_i^{t+1}) - \nabla P_i(\tilde{w}^*) \rangle \quad \text{(strong convexity of } P_i\text{)} \\
&= \langle u_i^{t+1} - \tilde{w}^*, -\lambda_i^{t+1} + \lambda_i^* + e_i^{t+1} \rangle \quad (\tilde{w}^* = u_i^*, \nabla P_i(u_i^*) = \lambda_i^*, \text{ and Eq. (53))} \\
&\leq \langle u_i^{t+1} - \tilde{w}^*, -\lambda_i^{t+1} + \lambda_i^* \rangle + \frac{\sigma}{2}\|u_i^{t+1} - \tilde{w}^*\|^2 + \frac{1}{2\sigma}\|e_i^{t+1}\|^2, \\
&\qquad (\langle a, b \rangle \leq t/2\|a\|^2 + 1/(2t)\|b\|^2 \text{ for all } a, b \in \mathbb{R}^d \text{ and } t > 0),
\end{aligned}$$

and

$$\begin{aligned}
\sigma \|w^{t+1} - \tilde{w}^*\|^2 &\leq \langle w^{t+1} - \tilde{w}^*, \nabla P_0(w^{t+1}) + h^{t+1} - \nabla P_0(\tilde{w}^*) - h^* \rangle \quad \text{(strong convexity of } P_0 + h\text{)} \\
&= \langle w^{t+1} - \tilde{w}^*, \sum_{i=1}^{n}[\lambda_i^{t+1} - \lambda_i^* - \rho_i(u_i^{t+1} - u_i^t)] + e_0^{t+1} \rangle, \quad \text{(Eq. (52) and Eq. (54))} \\
&\leq \langle w^{t+1} - \tilde{w}^*, \sum_{i=1}^{n}[\lambda_i^{t+1} - \lambda_i^* - \rho_i(u_i^{t+1} - u_i^t)] \rangle + \frac{\sigma}{2}\|w^{t+1} - \tilde{w}^*\|^2 + \frac{1}{2\sigma}\|e_0^{t+1}\|^2, \\
&\qquad (\langle a, b \rangle \leq t/2\|a\|^2 + 1/(2t)\|b\|^2 \text{ for all } a, b \in \mathbb{R}^d \text{ and } t > 0).
\end{aligned}$$

Summing up these inequalities and rearranging the terms, we obtain that

$$\frac{\sigma}{2}(\|w^{t+1} - \tilde{w}^*\|^2 + \sum_{i=1}^{n}\|u_i^{t+1} - \tilde{w}^*\|^2)$$

$$\leq \sum_{i=1}^{n}\langle w^{t+1} - \tilde{w}^*, \lambda_i^{t+1} - \lambda_i^* - \rho_i(u_i^{t+1} - u_i^t) \rangle + \frac{1}{2\sigma}\|e_0^{t+1}\|^2 + \sum_{i=1}^{n}(\langle u_i^{t+1} - \tilde{w}^*, -\lambda_i^{t+1} + \lambda_i^* \rangle + \frac{1}{2\sigma}\|e_i^{t+1}\|^2)$$

$$\leq \sum_{i=1}^{n}\langle w^{t+1} - u_i^{t+1}, \lambda_i^{t+1} - \lambda_i^* \rangle + \sum_{i=1}^{n}\rho_i\langle w^{t+1} - \tilde{w}^*, u_i^t - u_i^{t+1} \rangle + \frac{n+1}{2\sigma}\varepsilon_{t+1}^2$$

$$\qquad (\|e_i^{t+1}\| \leq \varepsilon_{t+1} \text{ for all } 0 \leq i \leq n \text{ and } t \geq 0)$$

$$\overset{Eq.\ (20)}{=} \sum_{i=1}^{n}\frac{1}{\rho_i}\langle \lambda_i^t - \lambda_i^{t+1}, \lambda_i^{t+1} - \lambda_i^* \rangle + \sum_{i=1}^{n}\rho_i\langle w^{t+1} - \tilde{w}^*, u_i^t - u_i^{t+1} \rangle + \frac{n+1}{2\sigma}\varepsilon_{t+1}^2, \tag{55}$$

Notice that the following well-known identities hold:

$$\langle w^{t+1} - \tilde{w}^*, u_i^t - u_i^{t+1} \rangle = \frac{1}{2}(\|w^{t+1} - u_i^{t+1}\|^2 - \|w^{t+1} - u_i^t\|^2 + \|\tilde{w}^* - u_i^t\|^2 - \|\tilde{w}^* - u_i^{t+1}\|^2), \tag{56}$$

$$\langle \lambda_i^t - \lambda_i^{t+1}, \lambda_i^{t+1} - \lambda_i^* \rangle = \frac{1}{2}(\|\lambda_i^* - \lambda_i^t\|^2 - \|\lambda_i^* - \lambda_i^{t+1}\|^2 - \|\lambda_i^t - \lambda_i^{t+1}\|^2). \tag{57}$$

These along with Eqs. (20) and (55) imply that

$$\frac{\sigma}{2}(\|w^{t+1} - \tilde{w}^*\|^2 + \sum_{i=1}^{n}\|u_i^{t+1} - \tilde{w}^*\|^2) + \sum_{i=1}^{n}\frac{\rho_i}{2}\|w^{t+1} - u_i^t\|^2 - \frac{n+1}{2\sigma}\varepsilon_{t+1}^2$$

$$\overset{Eq.\ (55))}{\leq} \sum_{i=1}^{n}\frac{1}{\rho_i}\langle \lambda_i^t - \lambda_i^{t+1}, \lambda_i^{t+1} - \lambda_i^* \rangle + \sum_{i=1}^{n}\rho_i\langle w^{t+1} - \tilde{w}^*, u_i^t - u_i^{t+1} \rangle + \sum_{i=1}^{n}\frac{\rho_i}{2}\|w^{t+1} - u_i^t\|^2$$

$$\overset{Eq.\ (56)}{\leq} \sum_{i=1}^{n}\frac{1}{\rho_i}\langle \lambda_i^t - \lambda_i^{t+1}, \lambda_i^{t+1} - \lambda_i^* \rangle + \sum_{i=1}^{n}\frac{\rho_i}{2}(\|\tilde{w}^* - u_i^t\|^2 - \|\tilde{w}^* - u_i^{t+1}\|^2 + \|w^{t+1} - u_i^{t+1}\|^2)$$

$$\overset{Eq.~(20)}{=} \sum_{i=1}^{n} \frac{1}{\rho_i}\langle \lambda_i^t - \lambda_i^{t+1}, \lambda_i^{t+1} - \lambda_i^* \rangle + \sum_{i=1}^{n} \frac{1}{2\rho_i}\|\lambda_i^{t+1} - \lambda_i^t\|^2 + \sum_{i=1}^{n} \frac{\rho_i}{2}(\|\tilde{w}^* - u_i^t\|^2 - \|\tilde{w}^* - u_i^{t+1}\|^2)$$

$$\overset{Eq.~(57)}{=} \sum_{i=1}^{n} \frac{1}{2\rho_i}(\|\lambda_i^* - \lambda_i^t\|^2 - \|\lambda_i^* - \lambda_i^{t+1}\|^2) + \sum_{i=1}^{n} \frac{\rho_i}{2}(\|\tilde{w}^* - u_i^t\|^2 - \|\tilde{w}^* - u_i^{t+1}\|^2)$$

$$= \sum_{i=1}^{n} [(\frac{\rho_i}{2}\|\tilde{w}^* - u_i^t\|^2 + \frac{1}{2\rho_i}\|\lambda_i^* - \lambda_i^t\|^2) - (\frac{\rho_i}{2}\|\tilde{w}^* - u_i^{t+1}\|^2 + \frac{1}{2\rho_i}\|\lambda_i^* - \lambda_i^{t+1}\|^2)]. \tag{58}$$

Summing up this inequality over $t = 0, \ldots, T$ for any $T \geq 0$, we obtain that

$$\sum_{t=0}^{T} \left[ \frac{\sigma}{2}\left(\|w^{t+1} - \tilde{w}^*\|^2 + \sum_{i=1}^{n}\|u_i^{t+1} - \tilde{w}^*\|^2\right) + \sum_{i=1}^{n} \frac{\rho_i}{2}\|w^{t+1} - u_i^t\|^2 - \frac{n+1}{2\sigma}\varepsilon_{t+1}^2 \right]$$

$$\leq \sum_{i=1}^{n} \left[ \left(\frac{\rho_i}{2}\|\tilde{w}^* - u_i^0\|^2 + \frac{1}{2\rho_i}\|\lambda_i^* - \lambda_i^0\|^2\right) - \left(\frac{\rho_i}{2}\|\tilde{w}^* - u_i^{T+1}\|^2 + \frac{1}{2\rho_i}\|\lambda_i^* - \lambda_i^{T+1}\|^2\right) \right]. \tag{59}$$

Recall from Algorithm 2 that $\varepsilon_{t+1} = q^t$, $u_i^0 = \tilde{w}^0$, and $\lambda_i^0 = -\nabla P_i(\tilde{w}^0)$. Notice from Eq. (54) that $\tilde{w}^* = u_i^*$ and $\lambda_i^* = -\nabla P_i(u_i^*)$. By these and Eq. (59), one can deduce that

$$\frac{\sigma}{2}(\|w^{t+1} - \tilde{w}^*\|^2 + \sum_{i=1}^{n}\|u_i^{t+1} - \tilde{w}^*\|^2) \leq \frac{n+1}{2\sigma}\sum_{t=0}^{\infty} q^{2t} + \sum_{i=1}^{n}\left(\frac{\rho_i}{2}\|\tilde{w}^* - u_i^0\|^2 + \frac{1}{2\rho_i}\|\lambda_i^* - \lambda_i^0\|^2\right)$$

$$\leq \frac{n+1}{2\sigma(1-q^2)} + \sum_{i=1}^{n}\left(\frac{\rho_i}{2}\|\tilde{w}^* - u_i^0\|^2 + \frac{1}{2\rho_i}\|\lambda_i^* - \lambda_i^0\|^2\right)$$

$$= \frac{n+1}{2\sigma(1-q^2)} + \sum_{i=1}^{n}\left(\frac{\rho_i}{2}\|\tilde{w}^* - \tilde{w}^0\|^2 + \frac{1}{2\rho_i}\|\nabla P_i(\tilde{w}^*) - \nabla P_i(\tilde{w}^0)\|^2\right).$$

In view of this and the definition of $\mathcal{Q}$ in Eq. (29), we can observe that $w^{t+1} \in \mathcal{Q}$ and $u_i^{t+1} \in \mathcal{Q}$ for all $t \geq 0$ and $1 \leq i \leq n$. Hence, the conclusion of this lemma holds as desired. $\qquad\square$

## B.4  Proof of Theorem 4.2

We first prove an auxiliary recurrence result that will be used later.

**Lemma B.1.** *Assume that $r, c > 0$ and $q \in (0, 1)$. Let $\{a_t\}_{t \geq 0}$ be a sequence satisfying*

$$(1+r)a_{t+1} \leq a_t + cq^{2t}, \quad \forall t \geq 0. \tag{60}$$

*Then we have*

$$a_{t+1} \leq \max\left\{q, \frac{1}{1+r}\right\}^{t+1}\left(a_0 + \frac{c}{1-q}\right), \quad \forall t \geq 0. \tag{61}$$

*Proof.* It follows Eq. (60) that

$$a_{t+1} \leq \frac{1}{1+r}a_t + \frac{1}{1+r}cq^{2t} \leq \frac{1}{(1+r)^2}a_{t-1} + \frac{cq^{2(t-1)}}{(1+r)^2} + \frac{cq^{2t}}{1+r}$$

$$\leq \cdots \leq \frac{1}{(1+r)^{t+1}}a_0 + \sum_{i=0}^{t} \frac{cq^{2i}}{(1+r)^{t+1-i}} = \frac{1}{(1+r)^{t+1}}a_0 + c\sum_{i=0}^{t} \frac{q^i}{(1+r)^{t+1-i}}q^i$$

$$\leq \frac{1}{(1+r)^{t+1}}a_0 + c\max\left\{q, \frac{1}{1+r}\right\}^{t+1}\sum_{i=0}^{t} q^i$$

$$(q^i \leq \max\{q, 1/(1+r)\}^i \text{ and } 1/(1+r)^{t+1-i} \leq \max\{q, 1/(1+r)\}^{t+1-i})$$

$$\leq \frac{1}{(1+r)^{t+1}} a_0 + \frac{c}{1-q} \max\left\{q, \frac{1}{1+r}\right\}^{t+1}$$

$$\leq \max\left\{q, \frac{1}{1+r}\right\}^{t+1} \left(a_0 + \frac{c}{1-q}\right).$$

Hence, Eq. (61) holds as desired. $\square$

The following lemma proves the Lipschitz continuity of $\nabla P_i$ on $\mathcal{Q}$.

**Lemma B.2.** *Let $\mathcal{Q}$ be defined in Eq. (29). Then there exists some $L_{\nabla P} > 0$ such that*

$$\|\nabla P_i(u) - \nabla P_i(v)\| \leq L_{\nabla P}\|u - v\|, \quad \forall u, v \in \mathcal{Q}, 0 \leq i \leq n. \tag{62}$$

*Proof.* Notice from Eq. (29) that the set $\mathcal{Q}$ is convex and compact. By this and the local Lipschitz continuity of $\nabla P_i$ on $\mathbb{R}^d$, one can verify that there exists some constant $L_{\nabla P} > 0$ such that Eq. (62) holds (see also Lemma 1 in Lu & Mei (2023)). $\square$

We introduce a potential function $S_t$ to measure the convergence of Algorithm 2:

$$S_t := \sum_{i=1}^{n}\left(\frac{\rho_i}{2}\|\tilde{w}^* - u_i^t\|^2 + \frac{1}{2\rho_i}\|\lambda_i^* - \lambda_i^t\|^2\right), \quad \forall t \geq 0. \tag{63}$$

The following lemma gives a recursive result of $S_t$, which will play a key role on establishing the global convergence rate for Algorithm 2 in Theorem 4.2.

**Lemma B.3.** *Suppose that Assumptions 0 to 2 hold. Let $\{w^{t+1}\}_{t\geq 0}$ and $\{u_i^{t+1}\}_{1\leq i\leq n, t\geq 0}$ be all the iterates generated by Algorithm 2. Then we have*

$$S_t \leq q_r^t\left[S_0 + \frac{1}{1-q}\left(\frac{n+1}{2\sigma} + \sum_{i=1}^{n}\frac{\sigma}{\rho_i^2 + 2L_{\nabla P}^2}\right)\right], \quad \forall t \geq 0, \tag{64}$$

*where $\sigma$ and $L_{\nabla P}$ are given in Eq. (17) and Lemma B.2, respectively, $q$ and $\rho_i$, $1 \leq i \leq n$, are inputs of Algorithm 2, and*

$$q_r := \max\left\{q, \frac{1}{1+r}\right\}, \quad r := \min_{1\leq i\leq n}\left\{\frac{\sigma\rho_i}{\rho_i^2 + 2L_{\nabla P}^2}\right\}. \tag{65}$$

*Proof.* Recall from Eq. (58) that

$$S_t = \sum_{i=1}^{n}\left(\frac{\rho_i}{2}\|\tilde{w}^* - u_i^t\|^2 + \frac{1}{2\rho_i}\|\lambda_i^* - \lambda_i^t\|^2\right)$$

$$\geq \sum_{i=1}^{n}\left(\frac{\rho_i + \sigma}{2}\|\tilde{w}^* - u_i^{t+1}\|^2 + \frac{1}{2\rho_i}\|\lambda_i^* - \lambda_i^{t+1}\|^2 + \frac{\rho_i}{2}\|w^{t+1} - u_i^t\|^2\right) + \frac{\sigma}{2}\|w^{t+1} - \tilde{w}^*\|^2 - \frac{n+1}{2\sigma}\varepsilon_{t+1}^2$$

$$\geq \sum_{i=1}^{n}\left(\frac{\rho_i + \sigma}{2}\|\tilde{w}^* - u_i^{t+1}\|^2 + \frac{1}{2\rho_i}\|\lambda_i^* - \lambda_i^{t+1}\|^2\right) - \frac{n+1}{2\sigma}\varepsilon_{t+1}^2. \tag{66}$$

Also, notice from Eqs. (53), (54) and (62) that

$$\|\lambda_i^* - \lambda_i^{t+1}\|^2 \overset{Eqs.\ (53)and\ (54)}{\leq} \left(\|\nabla P_i(\tilde{w}^*) - \nabla P_i(u_i^{t+1})\| + \|e_i^{t+1}\|\right)^2 \overset{Eq.\ (62)}{\leq} 2L_{\nabla P}^2\|\tilde{w}^* - u_i^{t+1}\|^2 + 2\varepsilon_{t+1}^2,$$

which implies that

$$\|\tilde{w}^* - u_i^{t+1}\|^2 \geq \frac{2\rho_i}{\rho_i^2 + 2L_{\nabla P}^2}\left(\frac{\rho_i}{2}\|\tilde{w}^* - u_i^{t+1}\|^2 + \frac{1}{2\rho_i}\|\lambda_i^* - \lambda_i^{t+1}\|^2\right) - \frac{2\varepsilon_{t+1}^2}{\rho_i^2 + 2L_{\nabla P}^2}. \tag{67}$$

Plugging this into Eq. (66), we have

$$S_t \overset{Eq.\ (67)}{\geq} \sum_{i=1}^{n} \left(1 + \frac{\sigma \rho_i}{\rho_i^2 + 2L_{\nabla P}^2}\right)\left(\frac{\rho_i}{2}\|\tilde{w}^* - u_i^{t+1}\|^2 + \frac{1}{2\rho_i}\|\lambda_i^* - \lambda_i^{t+1}\|^2\right) - \frac{n+1}{2\sigma}q^{2t} - \sum_{i=1}^{n}\frac{\sigma}{\rho_i^2 + 2L_{\nabla P}^2}q^{2t}$$

$$= (1+r)S_{t+1} - \left(\frac{n+1}{2\sigma} + \sum_{i=1}^{n}\frac{\sigma}{\rho_i^2 + 2L_{\nabla P}^2}\right)q^{2t} \qquad (r := \min_{1 \leq i \leq n}\sigma\rho_i/(\rho_i^2 + 2L_{\nabla P}^2)).$$

When $t = 0$, Eq. (64) holds clearly. When $t \geq 1$, by the above inequality, Eq. (65), and Lemma B.1 with $(a_t, c) = (S_t, \frac{n+1}{2\sigma} + \sum_{i=1}^{n}\frac{\sigma}{\rho_i^2 + 2L_{\nabla P}^2})$, we obtain that

$$S_t \leq \max\left\{q, \frac{1}{1+r}\right\}^t\left[S_0 + \frac{1}{1-q}\left(\frac{n+1}{2\sigma} + \sum_{i=1}^{n}\frac{\sigma}{\rho_i^2 + 2L_{\nabla P}^2}\right)\right]$$

$$= q_r^t\left[S_0 + \frac{1}{1-q}\left(\frac{n+1}{2\sigma} + \sum_{i=1}^{n}\frac{\sigma}{\rho_i^2 + 2L_{\nabla P}^2}\right)\right].$$

Therefore, the conclusion of this lemma is true as desired. □

*Proof of Theorem 4.2.* Notice that Algorithm 2 terminates when $\varepsilon_{t+1} + \sum_{i=1}^{n}\tilde{\varepsilon}_{i,t+1}$ is small. Next, we show that this quantity is bounded by $S_t$ defined in Eq. (63) plus other small quantities, and then use Lemma B.3 to bound the maximum number of iterations of Algorithm 2.

By Eq. (22), and the fact that $\|\nabla\varphi_{i,t}(u_i^{t+1})\|_\infty \leq \varepsilon_{t+1}$ (see Algorithm 2), one can obtain that

$$\varepsilon_{t+1} + \sum_{i=1}^{n}\tilde{\varepsilon}_{i,t+1} \overset{Eq.\ (22)}{=} \varepsilon_{t+1} + \sum_{i=1}^{n}\|[\nabla\varphi_{i,t}(w^{t+1}) - \rho_i(w^{t+1} - u_i^t)]\|_\infty$$

$$\leq \varepsilon_{t+1} + \sum_{i=1}^{n}\|\nabla\varphi_{i,t}(u_i^{t+1})\|_\infty + \sum_{i=1}^{n}\|\nabla\varphi_{i,t}(w^{t+1}) - \nabla\varphi_{i,t}(u_i^{t+1})\| + \sum_{i=1}^{n}\rho_i\|w^{t+1} - u_i^t\|$$

$$(\|u\|_\infty \leq \|u\| \text{ for all } u \in \mathbb{R}^d \text{ and the triangle inequality})$$

$$\leq (n+1)\varepsilon_{t+1} + \sum_{i=1}^{n}(L_{\nabla P} + \rho_i)\|w^{t+1} - u_i^{t+1}\| + \sum_{i=1}^{n}\rho_i\|w^{t+1} - u_i^t\|, \qquad (68)$$

where the second inequality follows from

$$\|\nabla\varphi_{i,t}(w^{t+1}) - \nabla\varphi_{i,t}(u_i^{t+1})\| \overset{Eq.\ (19)}{\leq} \|\nabla P_i(w^{t+1}) - \nabla P_i(u_i^{t+1})\| + \rho_i\|w^{t+1} - u_i^{t+1}\|$$

$$\overset{Eq.\ (62)}{\leq} (L_{\nabla P} + \rho_i)\|w^{t+1} - u_i^{t+1}\|, \qquad \forall 1 \leq i \leq n.$$

Next, we derive upper bounds for $\|w^{t+1} - u_i^{t+1}\|$ and $\rho_i\|w^{t+1} - u_i^t\|$, respectively. First, by Eqs. (58), (63) and (64), we have

$$\frac{\sigma}{4}\|w^{t+1} - u_i^{t+1}\|^2 \leq \frac{\sigma}{2}\|w^{t+1} - \tilde{w}^*\|^2 + \frac{\sigma}{2}\|u_i^{t+1} - \tilde{w}^*\|^2$$

$$\overset{Eq.\ (58)}{\leq} \sum_{i=1}^{n}(\frac{\rho_i}{2}\|\tilde{w}^* - u_i^t\|^2 + \frac{1}{2\rho_i}\|\lambda_i^* - \lambda_i^t\|^2) + \frac{n+1}{2\sigma}\varepsilon_{t+1}^2 \qquad (69)$$

$$= S_t + \frac{n+1}{2\sigma}q^{2t} \qquad (\text{the definition of } S_t \text{ in Eq. (63) and } \varepsilon_{t+1} = q^t \text{ for all } t \geq 0)$$

$$\overset{Eq.\ (64)}{\leq} q_r^t\left[S_0 + \frac{1}{1-q}\left(\frac{n+1}{2\sigma} + \sum_{i=1}^{n}\frac{\sigma}{\rho_i^2 + 2L_{\nabla P}^2}\right)\right] + \frac{n+1}{2\sigma}q^{2t}$$

$$\leq \left\{ q_r^{t/2} \left[ S_0 + \frac{1}{1-q} \left( \frac{n+1}{2\sigma} + \sum_{i=1}^{n} \frac{\sigma}{\rho_i^2 + 2L_{\nabla P}^2} \right) \right]^{1/2} + \sqrt{\frac{n+1}{2\sigma}} q^t \right\}^2$$

$$(a^2 + b^2 \leq (a+b)^2 \text{ for all } a, b \geq 0). \qquad (70)$$

Using again Eqs. (58), (63) and (64), we obtain that

$$\frac{1}{2}(\sum_{i=1}^{n} \rho_i \|w^{t+1} - u_i^t\|)^2 \leq (\sum_{i=1}^{n} \rho_i)(\sum_{i=1}^{n} \frac{\rho_i}{2} \|w^{t+1} - u_i^t\|^2) \qquad \text{(Cauchy-Schwarz inequality)}$$

$$\overset{Eq.\ (58)}{\leq} (\sum_{i=1}^{n} \rho_i) \sum_{i=1}^{n} (\frac{\rho_i}{2} \|\tilde{w}^* - u_i^t\|^2 + \frac{1}{2\rho_i} \|\lambda_i^* - \lambda_i^t\|^2) + (\sum_{i=1}^{n} \rho_i) \frac{n+1}{2\sigma} \varepsilon_{t+1}^2$$

$$= (\sum_{i=1}^{n} \rho_i) \left( S_t + \frac{n+1}{2\sigma} q^{2t} \right) \qquad \text{(the definition of } S_t \text{ in Eq. (63) and } \varepsilon_{t+1} = q^t \text{ for all } t \geq 0)$$

$$\overset{Eq.\ (64)}{\leq} (\sum_{i=1}^{n} \rho_i) \left\{ \left[ S_0 + \frac{1}{1-q} \left( \frac{n+1}{2\sigma} + \sum_{i=1}^{n} \frac{\sigma}{\rho_i^2 + 2L_{\nabla P}^2} \right) \right] q^t + \frac{n+1}{2\sigma} q^{2t} \right\}$$

$$\leq (\sum_{i=1}^{n} \rho_i) \left\{ q_r^{t/2} \left[ S_0 + \frac{1}{1-q} \left( \frac{n+1}{2\sigma} + \sum_{i=1}^{n} \frac{\sigma}{\rho_i^2 + 2L_{\nabla P}^2} \right) \right]^{1/2} + \sqrt{\frac{n+1}{2\sigma}} q^t \right\}^2$$

$$(a^2 + b^2 \leq (a+b)^2 \text{ for all } a, b \geq 0). \qquad (71)$$

Combining Eqs. (68), (70) and (71), we obtain that

$$\varepsilon_{t+1} + \sum_{i=1}^{n} \tilde{\varepsilon}_{i,t+1} \leq (n+1)q^t + \left( \frac{2}{\sqrt{\sigma}} \sum_{i=1}^{n} (L_{\nabla P} + \rho_i) + \sqrt{2 \sum_{i=1}^{n} \rho_i} \right)$$

$$\cdot \left\{ \left[ S_0 + \frac{1}{1-q} \left( \frac{n+1}{2\sigma} + \sum_{i=1}^{n} \frac{\sigma}{\rho_i^2 + 2L_{\nabla P}^2} \right) \right]^{1/2} q_r^{t/2} + \sqrt{\frac{n+1}{2\sigma}} q^t \right\}$$

$$\leq (n+1)q_r^{t/2} + \left( \frac{2}{\sqrt{\sigma}} \sum_{i=1}^{n} (L_{\nabla P} + \rho_i) + \sqrt{2 \sum_{i=1}^{n} \rho_i} \right)$$

$$\cdot \left\{ \left[ S_0 + \frac{1}{1-q} \left( \frac{n+1}{2\sigma} + \sum_{i=1}^{n} \frac{\sigma}{\rho_i^2 + 2L_{\nabla P}^2} \right) \right]^{1/2} + \sqrt{\frac{n+1}{2\sigma}} \right\} q_r^{t/2}$$

$$(q \leq q_r \leq q_r^{1/2} < 1). \qquad (72)$$

Recall from Algorithm 2 and Eq. (54) that $(u_i^0, \lambda_i^0) = (\tilde{w}^0, -\nabla P_i(\tilde{w}^0))$ and $\lambda_i^* = -\nabla P_i(\tilde{w}^*)$. By these and Eq. (63), one has

$$S_0 = \sum_{i=1}^{n} \left( \frac{\rho_i}{2} \|\tilde{w}^* - \tilde{w}^0\|^2 + \frac{1}{2\rho_i} \|\nabla P_i(\tilde{w}^*) - \nabla P_i(\tilde{w}^0)\|^2 \right). \qquad (73)$$

For convenience, denote

$$b := \left( \frac{2}{\sqrt{\sigma}} \sum_{i=1}^{n} (L_{\nabla P} + \rho_i) + \sqrt{2 \sum_{i=1}^{n} \rho_i} \right)$$

$$\cdot \left\{ \left[ \sum_{i=1}^{n} \left( \frac{\rho_i}{2} \|\tilde{w}^* - \tilde{w}^0\|^2 + \frac{1}{2\rho_i} \|\nabla P_i(\tilde{w}^*) - \nabla P_i(\tilde{w}^0)\|^2 \right) + \frac{1}{1-q} \left( \frac{n+1}{2\sigma} + \sum_{i=1}^{n} \frac{\sigma}{\rho_i^2 + 2L_{\nabla P}^2} \right) \right]^{1/2}$$

$$+ \sqrt{\frac{n+1}{2\sigma}} \right\}.$$

Using this, Eqs. (72) and (73), we obtain that

$$\varepsilon_{t+1} + \sum_{i=1}^{n} \tilde{\varepsilon}_{i,t+1} \leq (n+1+b)q_r^{t/2}.$$

This along with the termination criterion in Eq. (23) implies that the number of iterations of Algorithm 2 is bounded above by

$$\left\lceil \frac{2\log(\tau/(n+1+b))}{\log q_r} \right\rceil + 1 = \mathcal{O}(|\log \tau|). \tag{74}$$

Hence, the conclusion of this theorem holds as desired. □

We observe from the proof of Theorem 4.2 that under Assumptions 0 to 2, the number of iterations of Algorithm 2 is bounded by the quantity in Eq. (74).

## C   Proof of Theorem 3.2(b)

To establish the total inner-iteration complexity, we first show that all the inner iterates produced by Algorithm 2 for solving all the subproblems of form Eq. (12) within Algorithm 1 are in a compact set (i.e., $\mathcal{Q}_2$ later), and then estimate the Lipschitz modulus of $\nabla P_{i,k}$ for all $0 \leq i \leq n$ and all $k$ over $\mathcal{Q}_2$. Then we can bound the inner-iteration complexity based on the size of $\mathcal{Q}_2$ and the Lipschitz modulus.

**Boundedness of all inner iterates**   Recall from Eq. (15) that $\mathcal{Q}_1$ is a compact set. Let

$$U_{\nabla f} := \sup_{w \in \mathcal{Q}_1} \max_{1 \leq i \leq n} \|\nabla f_i(w)\|, \quad U_{\nabla c} := \sup_{w \in \mathcal{Q}_1} \max_{0 \leq i \leq n} \|\nabla c_i(w)\|, \quad U_c := \sup_{w \in \mathcal{Q}_1} \max_{0 \leq i \leq n} \|c_i(w)\|. \tag{75}$$

Since $\nabla f_i$ for all $1 \leq i \leq n$ are locally Lipschitz, they are Lipschitz on $\mathcal{Q}_1$ with some $L_{\nabla f} > 0$. Similarly, $\nabla c_i$ for all $0 \leq i \leq n$ are Lispchitz on $\mathcal{Q}_1$ with some modulus $L_{\nabla c} > 0$. Hence, $\nabla P_{i,k}$ for all $0 \leq i \leq n$ and all $k$ are Lipschitz continuous on $\mathcal{Q}_1$:

$$
\begin{aligned}
\|\nabla P_{i,k}(u) - \nabla P_{i,k}(v)\| &\overset{Eq. (50)}{\leq} \|\nabla f_i(u) - \nabla f_i(v)\| + \|[\mu_i^k + \beta c_i(u)]_+\| \|\nabla c_i(u) - \nabla c_i(v)\| \\
&\qquad + \|[\mu_i^k + \beta c_i(u)]_+ - [\mu_i^k + \beta c_i(v)]_+\| \|\nabla c_i(v)\| + \frac{1}{(n+1)\beta}\|u - v\| \\
&\overset{Eq. (81)}{\leq} L_{\nabla f}\|u-v\| + (\|\mu_i^*\| + \|\mu_i^k - \mu_i^*\| + \beta U_c)L_{\nabla c}\|u-v\| \\
&\qquad + \beta U_{\nabla c}^2\|u-v\| + \frac{1}{(n+1)\beta}\|u-v\| \\
&\overset{Eq. (47)}{\leq} L_{\nabla P,1}\|u-v\|
\end{aligned}
\tag{76}
$$

where

$$L_{\nabla P,1} := L_{\nabla f} + (\|\mu^*\| + r_0 + 2\sqrt{n}\bar{s}\beta + \beta U_c)L_{\nabla c} + \beta U_{\nabla c}^2 + \frac{1}{(n+1)\beta}. \tag{77}$$

The next lemma says that all the inner iterates generated by Algorithm 2 stay in a compact set.

**Lemma C.1.** *Suppose that Assumptions 0 to 2 hold and let $\{w^{k,t+1}\}_{t\geq 0}$ and $\{u_i^{k,t+1}\}_{1 \leq i \leq n, t \geq 0}$ be all the inner iterates generated by Algorithm 2 for solving the subproblems of form Eq. (12) in Algorithm 1. Then it holds that all these iterates stay in a compact set $\mathcal{Q}_2$, where*

$$\mathcal{Q}_2 := \left\{ v : \|v - u\|^2 \leq \frac{(n+1)^3\beta^2}{(1-q^2)} + (n+1)\beta \sum_{i=1}^{n} \left[ \left(\rho_i + \frac{L_{\nabla P,1}^2}{\rho_i}\right)(r_0 + 2\sqrt{n}\bar{s}\beta)^2 \right], u \in \mathcal{Q}_1 \right\}, \tag{78}$$

*and $L_{\nabla P,1}$ and $\mathcal{Q}_1$ are as defined in Eq. (77) and Eq. (15), respectively.*

*Proof.* Recall that for any $k \geq 0$, the subproblem in Eq. (12) has an optimal solution $w_*^k$ (see Eq. (40)), the initial iterate of Algorithm 2 for solving Eq. (12) is $w^k$, and $P_{i,k}$, $0 \leq i \leq n$, are strongly convex with modulus $1/[(n+1)\beta]$. By Lemma 4.2 with $(P_i, \tilde{w}^*, \tilde{w}^0, \sigma) = (P_{i,k}, w_*^k, w^k, 1/[(n+1)\beta])$, we obtain that $\{w^{k,t+1}\}_{t \geq 0}$ and $\{u_i^{k,t+1}\}_{t \geq 0, 1 \leq i \leq n}$ stay in a set $\widetilde{\mathcal{Q}}$ defined as

$$\widetilde{\mathcal{Q}} := \left\{ v : \|v - w_*^k\|^2 \leq \frac{(n+1)^3\beta^2}{(1-q^2)} + (n+1)\beta \sum_{i=1}^n \left( \rho_i \|w_*^k - w^k\|^2 + \frac{1}{\rho_i} \|\nabla P_{i,k}(w_*^k) - \nabla P_{i,k}(w^k)\|^2 \right) \right\}.$$

Thus, to show the boundedness of all the inner iterates, it suffices to derive upper bounds for $\|w_*^k - w^k\|$ and $\|\nabla P_{i,k}(w_*^k) - \nabla P_{i,k}(w^k)\|$ that are independent of $k$. Since $w^k, w_*^k \in \mathcal{Q}_1$, we have

$$\|\nabla P_{i,k}(w_*^k) - \nabla P_{i,k}(w^k)\| \leq L_{\nabla P,1} \|w_*^k - w^k\| \tag{79}$$

due to Eq. (76), where we note that $L_{\nabla P,1}$ is independent of $k$. Moreover, $\|w_*^k - w^k\| \leq r_0 + 2\sqrt{n}\bar{s}\beta$ from Eq. (47) provides a $k$-independent upper bound for $\|w_*^k - w^k\|$. Thus, we have

$$\sum_{i=1}^n \left( \rho_i \|w_*^k - w^k\|^2 + \frac{1}{\rho_i} \|\nabla P_{i,k}(w_*^k) - \nabla P_{i,k}(w^k)\|^2 \right) \leq \sum_{i=1}^n \left[ \left( \rho_i + \frac{L_{\nabla P,1}^2}{\rho_i} \right) (r_0 + 2\sqrt{n}\bar{s}\beta)^2 \right]. \tag{80}$$

Finally, combining the above results and noting that $w_*^k \in \mathcal{Q}_1$ completes the proof. $\qquad\square$

**Lipschitz modulus of $\nabla P_{i,k}$ for all $0 \leq i \leq n$ and all $k$ over $\mathcal{Q}_2$**   Let $L_{\nabla f,2}$ be the Lipschitz constant of $\nabla f_i$, $1 \leq i \leq n$, on $\mathcal{Q}_2$, and $L_{\nabla c,2}$ be the Lipschitz constant of $\nabla c_i$, $0 \leq i \leq n$, on $\mathcal{Q}_2$. Also, define

$$U_{\nabla c,2} := \sup_{w \in \mathcal{Q}_2} \max_{0 \leq i \leq n} \|\nabla c_i(w)\|, \qquad U_{c,2} := \sup_{w \in \mathcal{Q}_2} \max_{0 \leq i \leq n} \|c_i(w)\|. \tag{81}$$

Using similar arguments as for deriving $L_{\nabla P,1}$ in Eq. (77), we can see that $\nabla P_{i,k}$, $0 \leq i \leq n$, are Lipschitz continuous on $\mathcal{Q}_2$ with modulus $L_{\nabla P,2}$ defined as

$$L_{\nabla P,2} := L_{\nabla f,2} + (\|\mu^*\| + r_0 + 2\sqrt{n}\bar{s}\beta + \beta U_{c,2})L_{\nabla c,2} + \beta U_{\nabla c,2}^2 + \frac{1}{(n+1)\beta}. \tag{82}$$

**Proof of Theorem 3.2(b)**   Recall that Theorem 4.2 has established the number of iterations of Algorithm 2 for solving each subproblem of Algorithm 1. In the rest of this proof, we derive an upper bound for the total number of inner iterations for solving all subproblems of Algorithm 1.

We see from Lemma C.1 that all iterates generated by Algorithm 2 for solving Eq. (12) lie in $\mathcal{Q}_2$. Also, $\nabla P_{i,k}$, $1 \leq i \leq n$, are $L_{\nabla P,2}$-Lipschitz continuous on $\mathcal{Q}_2$. Therefore, by Theorem 4.2 with $(\tau, P_i, \sigma, L_{\nabla P}, \tilde{w}^*, \tilde{w}^0) = (\tau_k, P_{i,k}, 1/[(n+1)\beta], L_{\nabla P,2}, w_*^k, w^k)$ and the discussion at the end of Appendix B.4, one has that the number of iterations of Algorithm 2 for solving Eq. (12) during the $k$-th outer loop is no more than

$$T_k := \left\lceil \frac{2\log(\tau_k/(n+1+b_k))}{\log \tilde{q}_r} \right\rceil + 1 \tag{83}$$

where

$$\tilde{q}_r := \max\left\{ q, \frac{1}{1+\tilde{r}} \right\}, \quad \tilde{r} := \min_{1 \leq i \leq n} \left\{ \frac{\rho_i}{(n+1)\beta(\rho_i^2 + 2L_{\nabla P,2}^2)} \right\},$$

$$b_k := \left( 2\sqrt{(n+1)\beta} \sum_{i=1}^n (L_{\nabla P,2} + \rho_i) + \sqrt{2(\sum_{i=1}^n \rho_i)} \right)$$

$$\cdot \left\{ \left[ \sum_{i=1}^n \left( \frac{\rho_i}{2} \|w_*^k - w^k\|^2 + \frac{1}{2\rho_i} \|\nabla P_{i,k}(w_*^k) - \nabla P_{i,k}(w^k)\|^2 \right) \right. \right.$$

$$+ \frac{1}{1-q} \left( \frac{(n+1)^2\beta}{2} + \frac{1}{(n+1)\beta} \sum_{i=1}^{n} \frac{1}{\rho_i^2 + 2L_{\nabla P,2}^2} \right) \Bigg]^{1/2} + (n+1)\sqrt{\frac{\beta}{2}} \Bigg\}.$$

Plugging Eq. (80) into $\bar{b}$, we have that $b_k \leq \bar{b}$, where

$$\bar{b} := \left( 2\sqrt{(n+1)\beta} \sum_{i=1}^{n} (L_{\nabla P,2} + \rho_i) + \sqrt{2(\sum_{i=1}^{n} \rho_i)} \right)$$

$$\cdot \left\{ \left[ \sum_{i=1}^{n} \frac{\rho_i^2 + L_{\nabla P,1}^2}{2\rho_i}(r_0 + 2\sqrt{n}\bar{s}\beta)^2 + \frac{1}{1-q} \left( \frac{(n+1)^2\beta}{2} + \frac{1}{(n+1)\beta} \sum_{i=1}^{n} \frac{1}{\rho_i^2 + 2L_{\nabla P,2}^2} \right) \right]^{1/2} + (n+1)\sqrt{\frac{\beta}{2}} \right\},$$

which is independent of $k$. By $b_k \leq \bar{b}$, $\tau_k = \bar{s}/(k+1)^2$, $k \leq K_{\epsilon_1,\epsilon_2}$ where $K_{\epsilon_1,\epsilon_2}$ is the upper bound for the number of outer iterations as defined in Appendix A.3, and Eq. (83), one has that

$$T_k \leq \left\lceil \frac{2\log((n+1+\bar{b})(K_{\epsilon_1,\epsilon_2}+1)^2/\bar{s})}{\log(\tilde{q}_r^{-1})} \right\rceil + 1.$$

Therefore, by $K_{\epsilon_1,\epsilon_2} = \mathcal{O}(\max\{\epsilon_1^{-2}, \epsilon_2^{-2}\})$, one can see that the total number of inner iterations of Algorithm 1 is at most

$$\sum_{k=0}^{K_{\epsilon_1,\epsilon_2}} T_k \leq (K_{\epsilon_1,\epsilon_2}+1) \left( \left\lceil \frac{2\log((n+1+\bar{b})(K_{\epsilon_1,\epsilon_2}+1)^2/\bar{s})}{\log(\tilde{q}_r^{-1})} \right\rceil + 1 \right) = \widetilde{\mathcal{O}}(\max\{\epsilon_1^{-2}, \epsilon_2^{-2}\}). \tag{84}$$

Hence, Theorem 3.2(b) holds as desired.

---

**Algorithm 3** A centralized proximal AL method for solving Eq. (85)

---

**Input**: tolerances $\epsilon_1, \epsilon_2 \in (0,1)$, $w^0 \in \text{dom}(h)$, $\mu^0 \geq 0$, nondecreasing positive $\{\tau_k\}_{k \geq 0}$, and $\beta > 0$.

    **for** $k = 0, 1, 2, \ldots$ **do**

        Apply a centralized solver to find an approximate solution $w^{k+1}$ to:

$$\min_{w} \left\{ \ell_k(w) = f(w) + h(w) + \frac{1}{2\beta} \left( \|[\mu^k + \beta c(w)]_+\|^2 - \|\mu^k\|^2 \right) + \frac{1}{2\beta} \|w - w^k\|^2 \right\}$$

        such that

$$\text{dist}_\infty(0, \partial \ell_k(w^{k+1})) \leq \tau_k.$$

        Update the Lagrangian multiplier:

$$\mu^{k+1} = [\mu^k + \beta c(w^{k+1})]_+.$$

        Output $(w^{k+1}, \mu^{k+1})$ and terminate the algorithm if

$$\|w^{k+1} - w^k\|_\infty + \beta\tau_k \leq \beta\epsilon_1, \qquad \|\mu^{k+1} - \mu^k\|_\infty \leq \beta\epsilon_2.$$

    **end for**

---

# D  A centralized proximal AL method

In this part, we present a centralized proximal AL method (adapted from Algorithm 2 of Lu & Zhou (2023)) for solving the convex constrained optimization problem:

$$\min_{w} \ f(w) + h(w) \quad \text{s.t.} \quad c(w) \leq 0, \tag{85}$$

where the function $f : \mathbb{R}^d \to \mathbb{R}$ and the mapping $c : \mathbb{R}^d \to \mathbb{R}^m$ are continuous differentiable and convex, and $h$ is closed convex.

# E  Extra Numerical Results

## E.1  Dataset description for Neyman-Pearson classification

In this part, we describe the datasets for Neyman-Pearson classification in Section 5.1. 'breast-cancer-wisc', 'adult-a', and 'monks-1' are three binary classification datasets. We present the total number of samples for class 0 and class 1 and the number of features.

Table 3: Datasets for Neyman-Pearson classification

| dataset | class 0/class 1 | feature dimension |
|---|---|---|
| breast-cancer-wisc | 455/240 | 20 |
| adult-a | 24715/7840 | 21 |
| monks-1 | 275/275 | 21 |

## E.2  Linear equality constrained quadratic programming

In this subsection, we consider the linear equality constrained quadratic programming:

$$\min_{w} \sum_{i=1}^{n} \left( \frac{1}{2} w^T A_i w + b_i^T w \right) \quad \text{s.t.} \quad C_i w + d_i = 0, \quad 0 \le i \le n, \tag{86}$$

where $A_i \in \mathbb{R}^{d \times d}$, $1 \le i \le n$, are positive semidefinite, $b_i \in \mathbb{R}^d$, $1 \le i \le n$, $C_i \in \mathbb{R}^{\tilde{m} \times d}$, $0 \le i \le n$, and $d_i \in \mathbb{R}^{\tilde{m}}$, $0 \le i \le n$.

For each $(d, n, \tilde{m})$, we randomly generate an instance of Eq. (86). In particular, for each $1 \le i \le n$, we first generate a random matrix $A_i$ by letting $A_i = U_i D_i U_i^T$, where $D_i \in \mathbb{R}^{d \times d}$ is a diagonal matrix. The diagonal entries of $D_i$ are generated randomly from a uniform distribution over $[0.5, 1]$, and $U_i \in \mathbb{R}^{d \times d}$ is a randomly generated orthogonal matrix. We then randomly generate $C_i$, $0 \le i \le n$, with all entries drawn from a normal distribution with mean zero and a standard deviation of $1/\sqrt{d}$. Finally, we generate $b_i$, $1 \le i \le n$ and $d_i$, $0 \le i \le n$ as random vectors uniformly selected from the unit Euclidean sphere.

We apply Algorithm 1 and cProx-AL (Algorithm 3) to find a $(10^{-3}, 10^{-3})$-optimal solution of Eq. (86), and compare their solution quality. In particular, when implementing Algorithm 1, we exactly solve the quadratic subproblems in Eqs. (18) and (19). We run 10 trials of Algorithm 1 and cProx-AL, where for each run, both algorithms share the same initial point $w^0$, randomly chosen from the unit Euclidean sphere. We set the other parameters for Algorithm 1 and cProx-AL as $\mu_i^0 = (0, \dots, 0)^T \ \forall 0 \le i \le n$, $\bar{s} = 0.1$ and $\beta = 10$. We also set $\rho_i = 1$ for each $1 \le i \le n$ for Algorithm 2.

Table 4: Numerical results for Eq. (86) using our algorithm vs. using cProx-AL. Inside the parentheses are the respective standard deviations over 10 random trials.

| $n$ | $d$ | $\tilde{m}$ | objective value Algorithm 1 | cProx-AL | relative difference | feasibility violation Algorithm 1 | cProx-AL |
|---|---|---|---|---|---|---|---|
| | 100 | 1 | -0.23 (4.65e-6) | -0.23 (2.38e-5) | 1.63e-3 (1.01e-4) | 3.33e-4 (1.14e-5) | 5.68e-4 (2.82e-5) |
| 1 | 300 | 3 | -0.37 (2.74e-6) | -0.37 (1.32e-6) | 1.01e-3 (3.51e-5) | 3.52e-4 (1.44e-5) | 4.45e-4 (1.70e-5) |
| | 500 | 5 | -0.30 (1.36e-5) | -0.30 (7.54e-6) | 1.34e-3 (4.62e-5) | 4.38e-4 (5.36e-5) | 3.85e-4 (1.05e-5) |
| | 100 | 1 | 9.81 (7.18e-5) | 9.80 (1.46e-5) | 1.09e-3 (7.96e-6) | 1.34e-4 (9.02e-6) | 8.03e-4 (1.57e-6) |
| 5 | 300 | 3 | 8.47 (8.12e-5) | 8.45 (1.30e-5) | 1.36e-3 (9.62e-6) | 1.09e-4 (1.31e-5) | 8.28e-4 (1.98e-6) |
| | 500 | 5 | 9.92 (4.43e-5) | 9.91 (4.87e-6) | 8.26e-4 (4.27e-6) | 1.33e-4 (9.68e-6) | 3.73e-4 (2.43e-7) |
| | 100 | 1 | 49.40 (9.02e-5) | 49.37 (5.82e-6) | 5.59e-4 (1.67e-5) | 7.31e-5 (7.54e-6) | 5.88e-4 (1.34e-7) |
| 10 | 300 | 3 | 41.49 (7.04e-5) | 41.44 (5.48e-6) | 1.14e-3 (1.77e-6) | 8.56e-5 (2.27e-7) | 8.73e-4 (7.26e-6) |
| | 500 | 5 | 41.45 (2.25e-5) | 41.41 (5.30e-6) | 9.39e-4 (4.94e-7) | 9.29e-5 (2.55e-6) | 7.66e-4 (1.37e-7) |

We observe that Algorithm 1 and cProx-AL are capable of finding nearly feasible solutions, and achieve similar objective value. In view of the small standard deviations, we observe that the convergence behavior of Algorithm 1 remains stable across 10 trial runs.

