# OpenReview forum: "Federated Learning with Convex Global and Local Constraints"
_TMLR — Accepted by TMLR_

### Review · Reviewer_oLi8 · 2024-03-10

**Summary Of Contributions:**

The paper considers distributed and federated learning optimization problems. Unlike many previous works, the authors focus their attention on problems with constraints. In this setup, they developed a new method based on the alternating direction method of multipliers (ADMM), proved convergence guarantees, and ran experiments that support their claims.

**Audience:**

Yes

**Broader Impact Concerns:**

-

**Claims And Evidence:**

Yes

**Requested Changes:**

-

**Strengths And Weaknesses:**

I should acknowledge that I'm not from the ADMM community, so I can easily miss some important points. The problem considered is well-motivated and valid. Indeed, not many works analyze constrained optimization problems in the FL setting. The authors provided a good overview of the problem's current state. In general, the work looks promising and provides some insights to the ADMM community. I'll leave some questions that would help to understand the paper in more detail:
1. It is clear that the constrained distribution problem is a somewhat old problem. There should be many previous baselines and approaches that tackle this problem. While the authors discuss some of them in the related work section, I would like to see a more detailed comparison with previous baselines. For instance, the authors provide Theorem 3.2. It would be nice if they compare this complexity result with previous ones.
2. How do authors choose $\rho_i$? Why does the convergence rate not depend on these parameters? When the authors call Alg.2 in Alg.1, they should specify input parameters. The same question is to other parameters, including $\bar{s}, \dots$
3. Minor comment: Theorem 3.2 is slightly confusing. It provides complexity results for outer iterations and inner iterations of Alg.1, but I can see only one loop in Alg. 1. Is inner iterations of Alg.1 = iterations of Alg.2?

---

> ### Author Response · Authors · 2024-03-25
> **Thank you for the insightful comments!**
>
> > I should acknowledge that I'm not from the ADMM community, so I can easily miss some important points. The problem considered is well-motivated and valid. Indeed, not many works analyze constrained optimization problems in the FL setting. The authors provided a good overview of the problem's current state. In general, the work looks promising and provides some insights to the ADMM community.
>
> Thank you for your encouraging comments! This paper aims to tackle federated learning with constraints, an area that currently receives little research attention. As a byproduct of our development, our work also provides several new insights into the study of ADMM.
>
> > 1. It is clear that the constrained distribution problem is a somewhat old problem. There should
> be many previous baselines and approaches that tackle this problem. While the authors
> discuss some of them in the related work section, I would like to see a more detailed comparison
> with previous baselines. For instance, the authors provide Theorem 3.2. It would be nice if
> they compare this complexity result with previous ones
>
> We would like to reiterate the difference between distributed optimization/learning (DOL) and federated learning (FL): FL is a DOL framework that protects data privacy by **prohibiting the transfer of raw data from one client to another or to a central server**, whereas general DOL does not have this restriction. So, although FL can be viewed as a subset of DOL, general-purpose DOL algorithms may or may not be applicable for FL settings. We checked the existing DOL literature; the closest to our setting in Eq. (5) are Zhu & Mart ́ınez (2011) and Yuan et al. (2011), which do not violate the FL restriction and hence can be considered as FL algorithms, but they can only handle simple local constraints that are amenable to orthogonal projection. We will add these discussions to Section 1.3 in our revised manuscript.
>
> > 2. (a) How do authors choose $\rho_i$? Why does the convergence rate not depend on these parameters?
>
> From Eq. (18), $\rho_i$, $1\le i\le n$ can be viewed as weighting parameters for aggregation. Therefore, it is natural to set $\rho_i = a m_i$ for $1 \leq i \leq n$, where $m_i$ is the number of samples in client $i$ and $a$ is a global constant. We follow this rule when setting $\rho_i$'s.
>
> The iteration complexity of Algorithm 2 does depend on $\rho_i$'s, and other hyperparameters; the proofs in Appendix B.4 make the dependency explicit. In Theorem 2 of the main body, we use the big-O notation and only stress the dependency on $\epsilon_1$ and $\epsilon_2$, which is standard practice in complexity theory when people want to emphasize the convergence rate. We will highlight them below Theorem 3.2 in our revised manuscript.
>
> > 2. (b)  When the authors call Alg.2 in Alg.1, they should specify input parameters. The same question is to other parameters, including $\bar{s}$,...
>
> For Algorithm 1,
> 1. $\epsilon_1,\epsilon_2\in(0,1)$ only depend on the numerical accuracy that the user aims to achieve;
> 2. The initial iterates $w^0$ and $\mu^0_i$, $1\leq i \leq n$, are usually randomly generated or set as a constant vector;
> 3. $\bar{s}>0$ controls the tolerance sequence $\tau_k$, $k\ge0$ for the subproblems in Algorithm 1. These finite, non-zero tolerances allow us to solve the subproblems $\textbf{inexactly}$ but can still guarantee convergence, hence saving computational costs. In particular, setting $\tau_k$, $k\ge0$ to diminish rapidly towards zero at the order of $\mathcal{O}(1/k^2)$ can guarantee convergence of Algorithm 1. In practice, $\bar{s}$ only needs to be set as $\mathcal{O}(1)$.
>
> For Algorithm 2,
> 1. $(\tau,\tilde{w}^0)$ is specified as $(\tau_k,w^k)$ at the $k$th iteration of Algorithm 1.
> 2. $q\in(0,1)$ determines the tolerance sequence $\varepsilon_{t+1}$, $t\ge0$ for the subproblems in Eq. (18). These tolerances in solving subproblems reduce computational costs. Setting $\varepsilon_{t+1}$, $t\ge0$ to rapidly diminish toward zero rapidly at a geometric rate ensures the convergence of Algorithm 1. In practice, we suggest setting $q$ as $\mathcal{O}(1)$.
>
> We will add explanations about these algorithm hyperparameters to Sections 3.1 and 4.1 in our revised manuscript.
>
> > 3. Minor comment: Theorem 3.2 is slightly confusing. It provides complexity results for outer iterations and inner iterations of Alg.1, but I can see only one loop in Alg. 1. Is inner iterations of Alg.1 = iterations of Alg.2?
>
> Yes, the inner iterations are the iterations due to our call of Algorithm 2, i.e., the call above Eq (10), and the outer iterations are those contained in the for loop of Algorithm 1. The definitions of outer and inner iterations of Algorithm 1 are provided above Theorem 3.1. In this revision, to make this clear, we will turn on the line numbers for Algorithm 1, and state the scope of inner and outer iterations explicitly below Algorithm 1 (and above Theorem 3.1).

---

### Review · Reviewer_SQkm · 2024-03-12

**Summary Of Contributions:**

- The work considers a federated learning problem with local inequality constraints (that can only be evaluated locally) and additionally a global nonsmooth term and global inequality constraint
- They propose an inexact ADMM with verifiable termination criteria
- They show that local Lipschitz of the local objectives and constraints are sufficient for establishing convergence of their method under convexity

**Audience:**

Yes

**Claims And Evidence:**

Yes

**Requested Changes:**

Concerning lacking proper comparison to existing literature:

- Most of the construction and proofs seems to rely heavily on the centralized proximal AL of (Lu & Zhou 2023) which handles constraints. I suggest spending more time comparing up front and stating how you extend the results (i.e. local Lipschitz and federated setting)
- The application of inexact ADMM to federated learning has been proposed before (albeit without handling local constraints) [1,2,3,4] and the paper is not currently attributing proper credit.

My current understand is that the paper extends the results of (Lu & Zhou 2023) to a federated setting using a similar construction as [1], while relaxing the global Lipschitz condition to local Lipschitz. It would be very helpful to have this kind of comparison appear in the paper.


[1] https://arxiv.org/pdf/2204.10607.pdf

[2] https://www.ijcai.org/proceedings/2019/0629.pdf

[3] https://arxiv.org/pdf/2204.03529.pdf

[4] https://arxiv.org/pdf/2005.11418.pdf

----

Concerning overclaims:

The abstract claims that existing techniques "only deal with unconstrained FL problems". I would rephrase, since a proper comparison appears to be much more subtle than that. The work seems to specifically address the problem of _(private) local constraints for which the projection is not available or too expensive_. When projections are available constraints have been treated under federated composite optimization [5] and operator splitting approaches like Douglas-Rachford splitting [6,7,8].


Currently the text until page 2 suggests that the literature either does not handle constraints or at best treats very specific constraints such as class imbalance and fairness. Only under related work is it briefly mentioned that global and local constraints are treated but only when "simple".
I suggest:

- Specifying what is meant by "simple", since moving beyond "simple" is one of the main contributions. I would include a discussion as early as possible, but also after equation 5 would be a convenient place to address the difficulty of the local constraint considered.
- I would mention previous work on constraint problems much earlier.

[5] https://arxiv.org/pdf/2011.08474.pdf

[6] https://arxiv.org/pdf/2011.08474.pdf

[7] https://arxiv.org/pdf/2103.03452.pdf

[8] https://proceedings.neurips.cc/paper/2020/file/4ebd440d99504722d80de606ea8507da-Paper.pdf

-----

Minor:

- Appendix B.3 it is mentioned that "To the best of our knowledge, such boundedness results without assuming global Lipschitz continuity are entirely new in the literature on ADMM.?". It would be helpful to include references to local Lipschitz results in the centralized setting and explain what the main difficulty in extending it is.
- It might be misleading to write "establish its global linear convergence assuming strong convexity and locally Lipschitz continuous gradients." under contribution. If stepsizes are set appropriately my understanding is that the subproblem should be strongly convex when used as a subroutine in Algorithm 1. It would be good to mention that these assumptions are automatically satisfied under the convexity/smoothness assumption in Section 3.


----

I consider removing the overclaim important for recommending acceptance.

**Strengths And Weaknesses:**

Strengths:

- The paper is well-organized and the theoretical statements appears correct
- The proposed method handles local constraints that can only be accessed locally without relying on projections
- Relaxing global Lipschitz assumption to local Lipschitz appears to be new

Weaknesses:

- In its current form there are unnecessary overclaims concerning being first to address constrained problems in FL
- The writing currently lacks proper comparison with existing literature

See requested changes below for details.

---

> ### Author Response · Authors · 2024-04-01
> **Thank you for your insightful comments!**
>
> > 1. The work considers a federated learning problem with local inequality constraints (that can only be evaluated locally) and additionally a global nonsmooth term and global inequality constraint.
> 2. They propose an inexact ADMM with verifiable termination criteria.
> 3. They show that local Lipschitz of the local objectives and constraints are sufficient for establishing convergence of their method under convexity.
>
> > **Strengths:**
> > 1. The paper is well-organized and the theoretical statements appears correct.
> > 2. The proposed method handles local constraints that can only be accessed locally without relying on projections.
> > 3. Relaxing global Lipschitz assumption to local Lipschitz appears to be new.
>
> Thank you for your encouraging comments!
>
> > **Weaknesses:**
> > 1. In its current form there are unnecessary overclaims concerning being first to address constrained problems in FL.
> > 2. The writing currently lacks proper comparison with existing literature.
> > See requested changes below for details.
>
> Below, we will address your requested changes regarding overclaims and the literature comparison.
>
> > **Requested Changes:** Concerning lacking proper comparison to existing literature:
> > 1. Most of the construction and proofs seems to rely heavily on the centralized proximal AL of (Lu \& Zhou 2023) which handles constraints. I suggest spending more time comparing up front and stating how you extend the results (i.e. local Lipschitz and federated setting)
>
> Yes, our algorithmic ideas and proofs build on Lu \& Zhou (2023). Just to be clear, Lu \& Zhou (2023) proposes a proximal AL method for general constrained convex optimization, where the subproblems are solved by an accelerated gradient method. To make the proximal AL framework work in our FL setting, we have to change their algorithms as follows.
>
> 1. add communication steps to allow dual updates in an FL manner;
>
> 2. to solve the subproblem, we cannot directly apply the accelerated gradient method (AGM) as in Lu \& Zhou (2023). It is possible to develop an FL version of AGM by **eagerly** aggregating gradients from local clients, but that induces heavy communication between clients and the central server. To address this, we first reformulate the subproblem as a finite-sum problem and then propose an inexact ADMM solver to solve it. The inexact ADMM solver allows multiple steps of local updates before aggregation of model weights at the central server, hence it is communication friendly. We also propose a new stopping criterion for the inexact ADMM (Algorithm 2). Detailed explanations can be found in our response to the next question.
>
> To analyze the convergence of Algorithm 1, we establish the following new results:
>
> 1. modify the convergence analysis to adapt the algorithm modifications mentioned above;
> 2. analyze the FL algorithm under the assumption of locally Lipschitz continuous gradient.
>
> The complexity results of Algorithm 1 can be derived by adapting the analysis in Lu \& Zhou (2023), and there are no heavy technical obstacles for this adaptation.
>
> We will add these comparisons after Algorithm 1 in our revised manuscript.
>
> > 2. The application of inexact ADMM to federated learning has been proposed before (albeit without handling local constraints) [1,2,3,4] and the paper is not currently attributing proper credit.
>
> We would like to stress that the inexact ADMM is only used as a subproblem solver for our main algorithm, Algorithm 1. The main contribution of this paper is to propose a new algorithm to solve FL with constraints. Therefore, in the previous version of the paper, we focus on reviewing work related to the constrained FL problem in our ``related work" section (Section 1.3).
>
> However, we do agree that there are many ADMM-based algorithms proposed for FL or distributed optimization/learning. In particular, [1] proposes an ADMM-based FL algorithm that allows partial participation of the device, [3,4] develop ADMM-based FL algorithms to handle heterogeneity, and [2] applies ADMM to handle optimization problems with many constraints in a distributed manner. In this revision, we will include the discussion in the related work section.

---

> ### Author Response · Authors · 2024-04-01
> **Additional responses 1**
>
> > My current understand is that the paper extends the results of (Lu \& Zhou 2023) to a federated setting using a similar construction as [1], while relaxing the global Lipschitz condition to local Lipschitz. It would be very helpful to have this kind of comparison appear in the paper.
>
> > [1] https://arxiv.org/pdf/2204.10607.pdf
> > [2] https://www.ijcai.org/proceedings/2019/0629.pdf
> > [3] https://arxiv.org/pdf/2204.03529.pdf
> > [4] https://arxiv.org/pdf/2005.11418.pdf
>
> As said above, we have to make two important modifications to Lu \& Zhou (2023)'s centralized AL method to solve our constrained FL problem: (1) add communication steps to allow federated learning; (2) replace their accelerated-gradient-method-based subproblem solver by an inexact ADMM FL solver.
>
> We suppose that the current question is about the difference between the inexact ADMM FL solver in [1] and ours (**If not, please let us know**). The main innovations we have compared to [1] and other existing literature on FL include:
>
> 1. As the current reviewer points out, we establish the global linear convergence rate of an inexact ADMM under the assumption of locally Lipschitz continuous gradients, vs. globally Lipschitz continuous gradients in other work. To derive the convergence results under local Lipschitz conditions, we have constructed a bounded set that contains all the iterates generated by the local clients and the central server. Then, we leverage this boundedness result to extend the analysis of ADMM from the global Lipschitz condition to the local Lipschitz condition.
>
> 2. We propose a novel and rigorous stopping criterion that is easily verifiable, communication-light, and compatible with the outer iterations (as, again, our inexact ADMM FL algorithm serves as a subproblem solver in our overall algorithm framework). In contrast, since the ADMM algorithm in [1] is a standalone solver and does not consider integration with higher-level algorithms like our situation, so their stopping criterion is different from ours.
>
> We will add these discussions to Section 1.2 in this revision.
>
> > Concerning overclaims: The abstract claims that existing techniques ``only deal with unconstrained FL problems". I would rephrase, since a proper comparison appears to be much more subtle than that. The work seems to specifically address the problem of (private) local constraints for which the projection is not available or too expensive. When projections are available constraints have been treated under federated composite optimization [5] and operator splitting approaches like Douglas-Rachford splitting [6,7,8].
>
> Thank you for raising this very interesting point! Yes, our focus is on general convex constraints that may or may not be amenable to simple projections. But we totally agree that we should have discussed existing work such as [5,6,7,8] that work on FL with simple constraints. We will definitely follow this good suggestion, tune down our claims, and provide enough background on this.
>
> > Currently the text until page 2 suggests that the literature either does not handle constraints or at best treats very specific constraints such as class imbalance and fairness. Only under related work is it briefly mentioned that global and local constraints are treated but only when "simple". I suggest:
> > 1. Specifying what is meant by "simple", since moving beyond "simple" is one of the main contributions. I would include a discussion as early as possible, but also after equation 5 would be a convenient place to address the difficulty of the local constraint considered.
>
> Yes, as said, we will follow this suggestion in the revision.
>
>
> > 2. I would mention previous work on constraint problems much earlier.
>
> Yes, as said, we will follow this suggestion in the revision.
>
> > [5] https://arxiv.org/pdf/2011.08474.pdf
> > [6] https://arxiv.org/pdf/2011.08474.pdf
> > [7] https://arxiv.org/pdf/2103.03452.pdf
> > [8] https://proceedings.neurips.cc/paper/2020/file/4ebd440d99504722d80de606ea8507da-Paper.pdf

---

> ### Author Response · Authors · 2024-04-01
> **Additional responses 2**
>
> > **Minor:**
> > 1. Appendix B.3 it is mentioned that ``To the best of our knowledge, such boundedness results without assuming global Lipschitz continuity are entirely new in the literature on ADMM.?". It would be helpful to include references to local Lipschitz results in the centralized setting and explain what the main difficulty in extending it is.
>
> As far as we are aware, there is no existing complexity result on centralized ADMM under the assumption of local Lipschitz gradients.
>
> The general research on complexity analysis for optimization algorithms under local Lipschitz assumptions is relatively new. For example, Z. Lu, S. Mei (2023) proposes accelerated gradient methods for convex optimization problems with locally Lipschitz continuous gradients, and J. Zhang, M. Hong (2024) proposes accelerated gradient methods for nonconvex optimization problems with locally Lipschitz continuous gradients.
>
> [1] Z. Lu, S. Mei (2023) ``Accelerated first-order methods for convex optimization with locally Lipschitz continuous gradient"
>
> [2] J. Zhang, M. Hong (2024) ``First-order algorithms without Lipschitz gradient: A sequential local optimization approach"
>
> We will add references about algorithm developments under local Lipschitz continuity results in this revision.
>
> > 2. It might be misleading to write ``establish its global linear convergence assuming strong convexity and locally Lipschitz continuous gradients." under contribution. If stepsizes are set appropriately my understanding is that the subproblem should be strongly convex when used as a subroutine in Algorithm 1. It would be good to mention that these assumptions are automatically satisfied under the convexity/smoothness assumption in Section 3.
>
> Yes, that is right. In this revision, we will replace "establish its global linear convergence assuming strong convexity and locally Lipschitz continuous gradients" with "establish its global linear convergence for solving the subproblems of Algorithm 1, which are strongly convex and have locally Lipschitz continuous gradients."

---

> > ### Comment · Reviewer_SQkm · 2024-04-05
> >
> > I thank the authors for their thorough response. I've read the updated paper and find that it incorporates the missing comparisons and down-tones the claims. I don't see any issue in recommending accept.

---

### Review · Reviewer_VnvW · 2024-03-20

**Summary Of Contributions:**

The paper presents two algorithms for solving a Federated Learning (FL) problem under convex constraints. The authors motivate their approach with two practical applications and later demonstrate their efficiency through numerical simulations. Additionally, theoretical guarantees are provided to ensure the validity of their proposed methods.

**Audience:**

Yes

**Broader Impact Concerns:**

Including a discussion on potential societal impact does not seem relevant.

**Claims And Evidence:**

Yes

**Requested Changes:**

* The definition of a $(\epsilon_1,\epsilon_2)$-solution is interesting but differs from the classic measure of convergence for convex optimization algorithms. Can a broader discussion on the relevance of this metric be included?

* The result of Theorem 3.1 is based on the assumption: "If Algorithm 1 successfully terminates". But can we check if this assumption is valid? Furthermore, Lemma 3.1 shows that the iterates are bounded. Clarity could be improved with the addition of an explanation of the importance of this result for studying the convergence of Algorithm 1.

* Several parameters are introduced, such as $\rho_i, \epsilon_{i, t+1}, \epsilon_{t+1}$. A recommendation on tuning these parameters could improve the practical aspect of the algorithms. Is there an optimal choice for these parameters, and how should they be chosen with a fixed iteration and communication budget?

* [1] study the optimization of a similar FL problem (up to the Lagrangian dual transformation of Eq~5). Perhaps mentioning their work could be relevant.

[1] Konstantin Mishchenko, Grigory Malinovsky, Sebastian Stich, Peter Richt\'arik -- Proxskip: Yes! local gradient steps probably lead to communication acceleration! finally!

**Strengths And Weaknesses:**

**Strengths:**

+ The first two pages illustrate the significance of the investigated problem. Moreover, using the convergence metric inspired by Lu & Zhou (2023) adds originality to the results.

**Weaknesses:**

* The theoretical findings are not sufficiently exploited. The algorithm's parameters are not included in the results, and incorporating them could illustrate the influence of these parameters.

* Some sections are more difficult to follow, e.g., the introduction of Eq~(9). Perhaps the author should add more discussion at this stage.

---

> ### Author Response · Authors · 2024-03-26
> **Thank you for your insightful comments!**
>
> > The paper presents two algorithms for solving a Federated Learning (FL) problem under convex constraints. The authors motivate their approach with two practical applications and later demonstrate their efficiency through numerical simulations. Additionally, theoretical guarantees are provided to ensure the validity of their proposed methods.
>
> > **Strengths:** The first two pages illustrate the significance of the investigated problem. Moreover, using the convergence metric inspired by Lu & Zhou (2023) adds originality to the results.
>
> Thank you for your encouraging comments!
>
> > **Weaknesses:** 1. The theoretical findings are not sufficiently exploited. The algorithm's parameters are not included in the results, and incorporating them could illustrate the influence of these parameters.
>
> Below, we will address your requested changes regarding the theoretical findings and the algorithm's parameters.
>
> > 2. Some sections are more difficult to follow, e.g., the introduction of Eq. (9). Perhaps the author should add more discussion at this stage.
>
> We have moved some background technical materials to the appendix section due to page limit, e.g., Appendix D covers the basics of centralized AL method that is necessary for understanding Eq. (9). In this revision, we will make adjustment to make sure technical results are grounded and avoid surprises and confusion.
>
> > **Requested Changes:** 1. The definition of an $(\epsilon_1,\epsilon_2)$-solution is interesting but differs from the classical measure of convergence for convex optimization algorithms. Can a broader discussion on the relevance of this metric be included?
>
> For unconstrained convex problems with differentiable objective $\min_{w} f(w)$, a natural measure of convergence is $\|\|\nabla f(w)\|\|$, i.e., the distance between $0$ and $\nabla f(w)$, as the optimality condition is $\nabla f(w) = 0$. If the objective is nondifferentiable, we need to use the notation of subdifferential, $\partial f(w)$, which is a set for each $w$ in general. In this case, the optimality condition reads $0 \in \partial f(w)$, and the measure of convergence is the distance between $0$ and the subdifferent set $\mathrm{dist}(0, \partial f(w)) := \min_{u\in\partial f(w)}\|\|u\|\|$.
>
> Now for the constrained convex problem we consider
> $$
> \min_{w}\  f(w) + h(w) \quad \mathrm{s.t.}\ \ c(w)\le0,
> $$
> where $f$ is convex and continuously differentiable, $h$ is convex and nonsmooth, $c=[c_1,\ldots,c_m]^T$ with $c_i$ being convex and continuously differentiable, the optimality conditions is (described in the paper)
>
> $$(0,0)\in \begin{pmatrix}
>     \nabla f(w) + \partial h(w) + \nabla c(w) \mu, \\
>     c(w) - \mathcal{N}_{\mathbb{R}^m+}(\mu)
>     \end{pmatrix}.$$
>
> So a natural measure of convergence is $\mathrm{dist}(0, \nabla f(w)+\partial h(w)+\nabla c(w)\mu)$ and $\mathrm{dist}(0,c(w) - \mathcal{N}_{\mathbb{R}^m+}(\mu))$ =
>
> $\mathrm{dist}(c(w),\mathcal{N}_{\mathbb{R}^m+}(\mu))$, which leads to our definition of $(\epsilon_1,\epsilon_2)$-optimal solution in Definition 1.
>
> > 2. (a) The result of Theorem 3.1 is based on the assumption: ``If Algorithm 1 successfully terminates". But can we check if this assumption is valid?
>
> Theorem 3.1 is a partial, self-contained result, which we state to keep the proof for the master theorem, Theorem 3.2, clean. In the proof of Theorem 3.2, we show that this assumption always holds under Assumptions 0, 1 and 2 which are assumed throughout the whole paper.
>
> > 2. (b) Furthermore, Lemma 3.1 shows that the iterates are bounded. Clarity could be improved with the addition of an explanation of the importance of this result for studying the convergence of Algorithm 1.
>
> The boundedness result is essential for establishing the global convergence of Algorithm 1. In particular, our analysis only assumes local Lipschitz continuity on the gradient. The boundedness allows us to utilize the local Lipschitz continuity on a bounded set to establish the convergence rate for our proposed algorithm; see Appendices A.3 and C that contain proof of Theorem 3.2 for the details.
>
> We will add more explanations below Lemma 3.1 in our revised manuscript.

---

> ### Author Response · Authors · 2024-03-26
> **Additional responses**
>
> > 3. (a) Several parameters are introduced, such as $\rho_i,\epsilon_{i,t+1},\epsilon_{t+1}$. A recommendation on tuning these parameters could improve the practical aspect of the algorithms.
>
> For Algorithm 1,
> 1. $\epsilon_1,\epsilon_2\in(0,1)$ only depend on the numerical accuracy that the user aims to achieve;
> 2. The initial iterates $w^0$ and $\mu^0_i$, $1\leq i \leq n$, are usually randomly generated or set as a constant vector;
> 3. $\bar{s}>0$ controls the tolerance sequence $\tau_k$, $k\ge0$ for the subproblems in Algorithm 1. These finite, non-zero tolerances allow us to solve the subproblems $\textbf{inexactly}$ but can still guarantee convergence, hence saving computational costs. In particular, setting $\tau_k$, $k\ge0$ to diminish rapidly towards zero at the order of $\mathcal{O}(1/k^2)$, as we do in Algorithm 1, can guarantee convergence of Algorithm 1. In practice, $\bar{s}$ only needs to be set as $\mathcal{O}(1)$.
>
> For Algorithm 2,
> 1. $(\tau,\tilde{w}^0)$ is specified as $(\tau_k,w^k)$ at the $k$th iteration of Algorithm 1.
> 2. From Eq. (18), $\rho_i$, $1\le i\le n$ can be viewed as weighting parameters for aggregation. Therefore, it is natural to set $\rho_i = a m_i$ for $1 \leq i \leq n$, where $m_i$ is the number of samples in client $i$ and $a$ is a global constant. We follow this rule when setting $\rho_i$'s.
> 3. $q\in(0,1)$ determines the tolerance sequence $\varepsilon_{t+1}$, $t\ge0$ for the subproblems in Eq. (18). These tolerances in solving subproblems reduce computational costs. Setting $\varepsilon_{t+1}$, $t\ge0$ to rapidly diminish toward zero rapidly at a geometric rate ensures the convergence of Algorithm 1. In practice, we suggest setting $q$ as $\mathcal{O}(1)$.
>
> We will add explanations about these algorithm hyperparameters to Sections 3.1 and 4.1 in our revised manuscript.
>
> > 3. (b) Is there an optimal choice for these parameters, and how should they be chosen with a fixed iteration and communication budget?
>
> As explained above, it is hard to find an optimal choice for these parameters, but what matters most is their orders of magnitudes---which can be determined from their algorithmic roles as discussed above.
>
> Optimization with a fixed iteration and communication budget, especially stringent ones, is a very useful but challenging research topic. It is outside the scope of the current paper.
>
> We will add this as a future research direction in our revised manuscript.
>
> > 4. [1] study the optimization of a similar FL problem (up to the Lagrangian dual transformation of Eq.~(5)). Perhaps mentioning their work could be relevant.
> [1] Konstantin Mishchenko, Grigory Malinovsky, Sebastian Stich, Peter Richt'arik -- Proxskip: Yes! local gradient steps probably lead to communication acceleration! finally!
>
> The main result of [1] is applied to an unconstrained FL problem, vs. our constrained FL problem. We will add this reference to the related work in our revised manuscript.

---

### Review · Reviewer_TZnp · 2024-03-21

**Summary Of Contributions:**

This work studied federated learning with functional constraints in convex regimes. Byproximal augmented lagrangian, authors proposed an algorithm to solve such FL problems, and further analyzed its complexities. Numerical experiments result reveals comparable performances with respect to centralized algorithms.

**Audience:**

Yes

**Broader Impact Concerns:**

/

**Claims And Evidence:**

Yes

**Requested Changes:**

1. Typo:
   - Page 2, "... patient privacy (, OCR)..."
2. My search shows that there is a recent work which also studied Lagrangian method with applications in constrained FL: *Yao, Wei, et al. "Constrained Bi-Level Optimization: Proximal Lagrangian Value function Approach and Hessian-free Algorithm." arXiv preprint arXiv:2401.16164 (2024)* (in Appendix A.1.3, also they seems to be in bilevel problems which are more general than yours) or https://openreview.net/forum?id=xJ5N8qrEPl (more than 3 months earlier regarding their ICLR submission time stamp, but I agree the time relative to your TMLR submission is tricky). Would you mind commenting on it comparing to your work?

**Strengths And Weaknesses:**

Strength:
1. Detailed illustration on problem transformation to fit the federated structure
2. First work in FL with general constraints with a systematical study
3. The presentation is easy to follow.


Weakness:
1. I am a bit confused on your statement regarding you are the "first result" extending to local Lipschitz in related FL literature. Following your proof (e.g., Lemma B.2, Theorem 3.2(b)), the radius of the set $\mathcal{Q}$ depends on many parameters like the number of client $n$, and accuracy $q$, basically I think they can be arbitrarily large (say high accuracy, or large client size), and you ask the local Lipschitz to **hold in every possible $\mathcal{Q}\subseteq\mathbb{R}^d$ (i.e., $\mathcal{Q}_1$ and $\mathcal{Q}_2$)**. Also with the presence of $q$, I suspect possibly the corresponding Lipschitz constant $L_P$, can be very large, and I fear there is not a very significant difference compared to the common global Lipschitz assumption. Could you please provide more comments or insights on why such local Lipschitz is important?
2. Even though claimed to be the first work in constrained FL problems, the proof strategy relies on the work of Lu & Zhou (2023), also there lacks a detailed discussion on how to understand the results, for example, a discussion on the sample/communication complexity based on Theorem 3.2 (i.e., setting detailed values of $\epsilon_1$ and $\epsilon_2$) will be helpful.
3. Also you mentioned cProx-AL algorithm adapted from Lu & Zhou (2023), and applied it in the experiments. They derived the complexity is $O(\epsilon^{-1}\log\epsilon^{-1})$, I am wondering whether cProx-AL algorithm you mentioned also enjoy the rate in FL regime, and how do you compare the (theoretical) rates between your algorithm and cProx-AL.

---

> ### Author Response · Authors · 2024-03-29
> **Thank you for your insightful comments!**
>
> > This work studied federated learning with functional constraints in convex regimes. By proximal augmented lagrangian, authors proposed an algorithm to solve such FL problems, and further analyzed its complexities. Numerical experiments result reveals comparable performances with respect to centralized algorithms.
>
> > **Strengths:**
> > 1. Detailed illustration on problem transformation to fit the federated structure.
> > 2. First work in FL with general constraints with a systematical study.
> > 3. The presentation is easy to follow.
>
> Thank you for your encouraging comments!
> > **Weakness:**
> > 1. I am a bit confused on your statement regarding you are the ``first result" extending to local Lipschitz in related FL literature. Following your proof (e.g., Lemma B.2, Theorem 3.2(b)), the radius of the set $\mathcal{Q}$ depends on many parameters like the number of client $n$, and accuracy $q$, basically I think they can be arbitrarily large (say high accuracy, or large client size), and you ask the local Lipschitz to hold in every possible $\mathcal{Q}\subseteq\mathbb{R}^d$ (i.e., $\mathcal{Q}_1$ and
> $\mathcal{Q}_2$). Also with the presence of $q$, I suspect possibly the corresponding Lipschitz constant $L_P$, can be very large, and I fear there is not a very significant difference compared to the common global Lipschitz assumption. Could you please provide more comments or insights on why such local Lipschitz is important?
>
> We stress that the major contribution of this paper is the study of FL problems with general constraints. There is no previous result on this under whatever conditions. In particular, our result assumes local Lipschitz gradient, but there are no such results under global Lipschitz gradient conditions.
>
> When we say that a function $h(x)$ is locally Lipschitz, we mean that for every point $x$, there exists a neighborhood of $x$ so that $h$ has a finite Lipschitz constant. When $h$ is locally Lipschitz, over any compact set $\mathcal{Q}$, there is a finite Lipschitz constant over $\mathcal{Q}$.
>
> We agree that Lipschitz constants $L_{\nabla P,1}$ and $L_{\nabla P,2}$ for $\mathcal{Q}_1\text{ and }\mathcal{Q}_2$ depend on many parameters and can be large in practice.
>
> But, the dependencies of $L_{\nabla P,1}\text{ and }L_{\nabla P,2}$ on the various factors are all low-degree polynomials, which is reasonable. Qualitatively, local Lipschitz conditions are strictly weaker than global Lipschitz conditions, and the problem with assuming global Lipschitz conditions---which may not hold even if local Lipschitz conditions hold---is whether we can derive a convergence result at all.
>
> We will add more discussion to Sections 3.1 and 4.1 in this revision.
>
> > 2. (a) Even though claimed to be the first work in constrained FL problems, the proof strategy relies on the work of Lu \& Zhou (2023),
>
> Our algorithmic ideas and proofs build on Lu \& Zhou (2023), but we study the problem in a federated learning setting, vs. their centralized setting. So we have to modify the algorithms and corresponding proofs to make them work for our setting. We will add the discussion to Section 1.2 in this revision.
>
> > 2. (b) also there lacks a detailed discussion on how to understand the results, for example, a discussion on the sample/communication complexity based on Theorem 3.2 (i.e., setting detailed values of $\epsilon_1\text{ and }\epsilon_2$) will be helpful.
>
> Theorem 3.2 provides an upper bound on the outer-loop iteration complexity as well as the total iteration complexity. We can estimate the communication complexity as follows: since each iteration of Algorithms 1 and 2 requires one round of communication, the communication complexity of Algorithm 1 is $\widetilde{\mathcal{O}}(\max\\{\epsilon_1^{-2},\epsilon_2^{-2}\\})$ and the communication complexity of each call of Algorithm 2 is $\mathcal{O}(|\log\tau|)$. We will add these results to Sections 3.3 and 4.3 in this revision.
>
> The notion of sample complexity in machine learning typically concerns both training and generalization. Here, our focus is only on training and no generalization yet, and so we do not consider sample complexity here.
>
> > 3. Also you mentioned cProx-AL algorithm adapted from Lu \& Zhou (2023), and applied it in the experiments. They derived the complexity is $\mathcal{O}(\epsilon^{-1}\log\epsilon^{-1})$, I am wondering whether cProx-AL algorithm you mentioned also enjoy the rate in FL regime, and how do you compare the (theoretical) rates between your algorithm and cProx-AL.
>
> We would like to reiterate that cProx-AL is proposed in Lu \& Zhou (2023) for general constrained convex optimization in the **centralized** setting, while our main algorithm, Algorithm 1, tackles the **federated** setting. Since these two algorithms are tackling two different problem settings, it is not appropriate to directly compare their complexity results.

---

> ### Author Response · Authors · 2024-03-29
> **Additional responses**
>
> > **Requested Changes:** 1. Typo: Page 2, ``... patient privacy (,OCR) ..."
>
> We will correct this typo in this revision.
>
> > 2. My search shows that there is a recent work which also studied Lagrangian method with applications in constrained FL: Yao, Wei, et al. ``*Constrained Bi-Level Optimization: Proximal Lagrangian Value function Approach and Hessian-free Algorithm*." arXiv preprint arXiv:2401.16164 (2024) (in Appendix A.1.3, also they seems to be in bilevel problems which are more general than yours) or \url{https://openreview.net/forum?id=xJ5N8qrEPl} (more than 3 months earlier regarding their ICLR submission time stamp, but I agree the time relative to your TMLR submission is tricky). Would you mind commenting on it comparing to your work?
>
> We will focus on the lower-level constrained FL problem in their paper, which is relevant to our paper, and omit their bi-level optimization aspect: their lower-level problem contains simple constraints that come from variable splitting (i.e., consensus constraints) and can be handled easily by closed-form projections, whereas our focus is on general convex constraints that may or may not be amenable to simple projections. We will add this to our related work in this revision.

---

### Author Response · Authors · 2024-04-02
**The revised manuscript uploaded**

Thank you for taking the time and effort to review our paper and provide insightful comments. We have revised our manuscript according to your suggestions. The revised version has now been uploaded, with major changes highlighted in blue.

---

### Decision · Action_Editor_CQJj · 2024-04-25

**Recommendation:** Accept as is

**Comment:**

The authors have addressed most of the concerns raised by the reviewers. Therefore, the reviewers concur that the paper represents a valuable addition to the existing literature and recommend its acceptance. I am in agreement with this assessment.

**Audience:**

The paper provides an interesting contribution to the area of constrained optimization for federated learning, which is relevant to the TMLR audience.

**Claims And Evidence:**

This paper considers Federated Learning optimization problems with general constraints. Building upon previous work [1], the authors considers an algorithm based on a Lagragian Augmented approach. They establish theoretical guarantees under the conditions that the objective function and the function defining the constraints are convex and locally Lipschitz. Finally, they illustrate their method through classification and fairness-aware learning tasks.

[1] Zhaosong Lu and Zirui Zhou. Iteration-complexity of first-order augmented Lagrangian methods for convex conic programming. SIAM Journal on Optimization, 33(2):1159–1190, 2023.